# Negativeome characterization and decontamination in early-life virome studies

Nataliia Kuzub ⑩ , Alexander Kurilshikov ⑩ , Alexandra Zhernakova ⑩ & Sanzhima Garmaeva ⑩ ✉

Contaminant sequences of external origin complicate the study of host-associated viromes, particularly in low-biomass samples obtained through viral-like particle (VLP) enrichment. However, the prevalence and impact of external contaminants on low-biomass samples are under-studied. Here, we analyze 1321 gut virome samples and 55 negative controls (NCs) from four early-life virome studies. Virus sequences identified in NCs, termed negativeome, were used as a proxy for the contamination assessment. We show that 61% of samples share at least one identical strain with negativeome, likely representing external contamination. While the median abundance of contaminant strains in these samples is only 1%, it ranges from 0 to 99% and exceeds 10% in 11% of infant samples. We further demonstrate that contamination is largely study-specific and has a greater impact on infant samples than on maternal samples. Based on our results, we propose a contamination assessment method using a publicly available database of sequences detected in NCs and a strain-level decontamination strategy.

Viruses are the most abundant biological entities on Earth, and they constitute a significant component of the human gut microbiome. The number of viruses in the human gut has been estimated to exceed $10^{12}$, roughly equal to the number of bacteria[1]. Despite this quantity, the total weight of the human gut virome's genetic material can be just ~50 micrograms[1]. This makes virome extraction and annotation challenging, particularly for low-diversity samples like those from early life[2–5]. A major issue here is distinguishing genuine sample signals from external or environmental contamination[5,6], and this has been a topic of recent debate in relation to the microbiome of the placenta[7,8] and blood[9]. To address this issue, proper no-template controls or negative controls (NCs), processed alongside biological samples, are essential at every step of sample processing. Best practices include sequencing NCs alongside the samples and removing sequences identified in NCs during data analysis[2–5,10,11]. Previous studies have linked the environmental contamination identified in virome samples to various laboratory components used for nucleic acid extraction and sequencing, including individual reagents[6], entire extraction kits[12], and laboratory plastics[5]. However, the impact of contamination on virome samples, its

sharedness across different studies, and appropriate decontamination strategies remain under-studied.

Here, we aimed to assess the impact of environmental contamination on the low-biomass samples obtained using viral-like particle (VLP) enrichment protocols and elucidate the level of genomic resolution necessary for virome decontamination. To do so, we annotated the viral composition of NCs, termed the negativeome, and tracked the negativeome sequences in biological samples as a proxy for environmental contamination. We employed publicly available data from four early-life virome studies[2–4,13] that had used VLP enrichment protocols and deposited raw sequencing data for both biological samples and NCs in public archives. Together, these studies include 1321 biological samples (1175 infant samples from 0 to 5 years and 146 maternal samples) and 55 NCs (Supplementary Data 1 and Supplementary Fig. 1), in which we identified 971,583 putative virus sequences that clustered to 193,970 viral operational taxonomic units (vOTUs), providing a general representation of species-level viral clusters (see "Methods", Supplementary Fig. 2a, b).

Department of Genetics, University of Groningen, University Medical Center Groningen, Groningen, the Netherlands. ✉e-mail: sana.garmaeva@gmail.com

## Results

### NCs and samples cannot be reliably distinguished using key genomic and ecological features

We first aimed to determine if specific features (number of reads and reconstructed contigs, virus sequences, viral genome completeness, viral diversity and richness) differed between NCs and biological samples. While clean reads and Shannon diversity were similar between groups (p-value > 0.2, Supplementary Data 2 and 6, Fig. 1a, b, and Supplementary Fig. 3), the number of reconstructed contigs, viral sequences, viral genome completeness, and viral richness were generally lower in NCs compared to biological samples (p-value < 0.04, Supplementary Data 3−5 and 7, Fig. 1a, b, and Supplementary Fig. 3). However, both the direction and significance of associations varied across the studies for all features tested except the number of reconstructed viral sequences. For example, viral richness was significantly different between NCs and biological samples in all three tested studies (FDR <1e-02; Supplementary Data 6), but the direction of this difference was inconsistent (Fig. 1b). This inconsistency likely reflects both study-to-study variability in vOTU richness (intraclass correlation coefficient (ICC) = 0.2, Supplementary Data 8) and large variation among biological samples (median: 230 vOTUs, interquartile range (IQR): 64−594).

Next, we tested whether NCs could be distinguished from biological samples without prior knowledge using the genomic and ecological features that differed between NCs and biological samples (i.e., the number of reconstructed contigs and viral sequences, genome completeness, and richness). We hypothesized that if these features reliably capture biological signals, then a model constructed based on them to predict a quantifiable variable−sampling age−would assign implausible or outlier-like ages to NCs, which by definition do not have a biological age. To test this, we built a linear model to predict sampling age from these four features in two longitudinal studies. In the Liang et al. dataset, the inferred sampling age of NCs averaged 1.3 ± 0.6 months, which falls within the observed range for biological samples. Similarly, for the Garmaeva et al. dataset, the NC was assigned an age of 7.9 months, which is also within the dataset's observed range (1−12 months). These results suggest that the combined technical and ecological features, while differing in group comparisons, do not provide a strong enough signal to reliably separate NCs from biological samples.

We next investigated differences in virome composition between NCs and biological samples, categorizing vOTUs with ≥50% genome completeness by nucleic acid type (dsDNA, ssDNA, RNA) based on their assigned taxonomy. Here, we observed a significantly lower proportion of ssDNA viruses in NCs compared to samples across all studies (beta = −14.9, FDR = 1e-04, Supplementary Data 9), driven mainly by a significant difference in one of the studies (Liang et al., FDR = 2e-02). When further tested separately, samples from Shah et al. had fewer RNA viruses than NCs (beta = −1.6, FDR = 2.3e-05, Supplementary Fig. 4a and Supplementary Data 9).

At the level of the predicted host domain, prokaryotic viruses dominated all datasets (p-value = 2.4e-129, Supplementary Fig. 4b and Supplementary Data 10), consistent with previous findings[14,15], with no difference between NCs and samples (FDR > 0.1, Supplementary Data 11 and Supplementary Fig. 4b). Study-specific testing revealed a lower proportion of prokaryotic viruses in NCs compared to samples in one study (Shah et al., FDR = 2e-05). At the genus level of prokaryotic viruses' hosts, NCs primarily contained viruses with hosts typical of the human gut microbiota, such as *Alistipes*, *Bacteroides*, *Bifidobacterium*, and *Escherichia* (Supplementary Fig. 5).

We next assessed the similarity in composition between biological samples and NCs at the vOTU level (1−Bray−Curtis dissimilarity, "Methods"). Overall, unrelated biological samples showed a high degree of individual specificity and low inter-similarity, in agreement with previous studies[14–16] (Supplementary Fig. 6). In three of the four studies, the similarity between unrelated samples was greater than the similarity between samples and NCs (p-value < 0.0004). In the

remaining study, biological samples were more similar to NCs than to other biological samples (Supplementary Fig. 6, p-value < 0.0004, Supplementary Data 12). Overall, the similarity of NC composition to that of biological samples was low but comparable to that between unrelated biological samples. We also observed NCs clustering with the samples at both the vOTU level and the level of host-based vOTU aggregates (Fig. 1c, d and Supplementary Data 13). Notably, most of the NCs clustered with infant samples collected during the first four months of life, indicating that the early-life human gut virome exhibits similarity to the NC virome.

Overall, while the composition of NCs and biological samples differed, the variation between NCs and samples was similar to the variation observed between unrelated samples. Although we observed differences between NCs and biological samples in some genomic and ecological features, the direction and significance were often inconsistent across the studies. Additionally, we demonstrated that NCs could not be reliably distinguished from biological samples based on genomic and ecological features.

### Properties of NC-associated contaminants across the four early-life studies

We further investigated whether external viral contaminants varied across the datasets by comparing the viruses identified in NCs from different studies. In total, 5984 vOTUs were found in NCs from all four studies (Supplementary Fig. 7), with the majority (N = 5339) identified in the NCs from Shah et al. While no vOTUs were shared across all NCs, two vOTUs were found in NCs from three studies, 44 vOTUs were shared between NCs from two studies, and 5938 vOTUs were study-specific (Fig. 2a). At strain level (see "Methods"), only three viruses were shared between NCs and samples from two studies. Of note, two Caudoviricetes phages that were shared between two studies from the U.S[2,3]. are predicted to infect *Burkholderia cepacia* complex species (Fig. 2b), which are common environmental contaminants often shared across U.S. hospitals[11,17,18]. Another strain-level virus that was shared across two studies, phiX174, is commonly used as a positive sequencing control[5,19] (Fig. 2c).

We also compared the sequences detected in NCs to the genomes of viral taxa reported to be laboratory-component-associated (LCA) viral sequences[6] by Asplund et al. (2019). Of the 5984 NC sequences, only 34 (0.6%) matched the previously reported LCA viral sequences (see "Methods"). Of these, 60.1% (N = 20) belonged to Bordetella phage, EBPR podovirus, and sewage-associated circular DNA viruses (Supplementary Data 14). Among the 677 NC sequences with NT (non-redundant nucleotide database) viral taxonomic assignments that did not overlap with the LCA taxa, 529 were linked to metagenomically reconstructed viral genomes for which taxonomy was only resolved up to the class level. Interestingly, within the subset of sequences assigned to isolate viral genomes, we identified a few previously reported reagent-associated CRESS-like viruses[20] (Supplementary Data 15).

In line with the limited vOTU-sharing we observed across NCs from different datasets, we found that NC vOTU compositions were more similar within the same study than across different studies (FDR <3e-04, Fig. 2d and Supplementary Data 16). NCs also showed higher similarity to samples within their study than to samples from other studies (FDR <4e-04, Fig. 2d and Supplementary Data 16). While these results suggest that contamination is more likely to be study-specific, the overlap of some NC-detected viral species with previously reported contaminants and the presence of a few viral strains that were shared across studies reflect common environmental and technical contamination.

### Early-life gut virome samples are more susceptible to contamination

We then quantified sample contamination by analyzing the proportion and abundance of vOTUs shared with NCs within each study. Only 20.3% (N = 256) of all biological samples did not share any vOTUs with

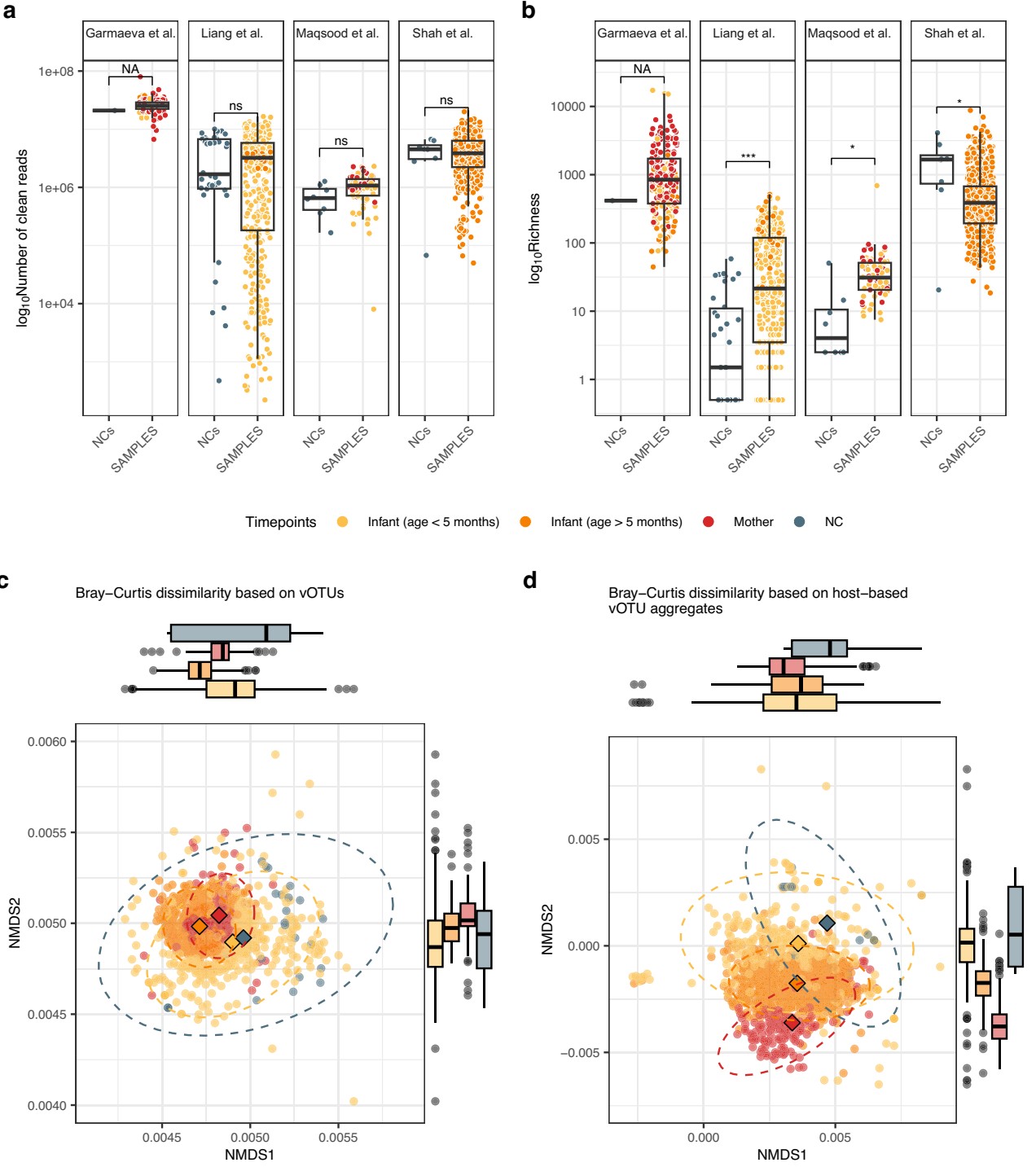

**Fig. 1 | Comparison of sequencing depth, richness, and viral community composition between negative controls and samples. a** Number of clean reads in negative controls (NCs) vs samples. **b** Viral operational taxonomic unit (vOTU) richness in NCs vs samples. In (**a**, **b**), data are shown for 1313 samples and 55 NCs, distributed by study as follows: Garmaeva et al. (NCs = 1, samples = 205); Liang et al. (NCs = 38, samples = 383); Maqsood et al. (NCs = 8, samples = 78); Shah et al. (NCs = 8, samples = 647). **c**, **d** Non-metric multidimensional scaling analysis utilizing Bray−Curtis dissimilarity, computed at: **c** the vOTU level; **d** predicted host level. In (**c**), data are shown for 33 NCs and 1228 samples: infants <5 months (*n* = 372), infants >5 months (*n* = 710), mothers (*n* = 146). In (**d**), data are shown for 25 NCs and

1147 samples: infants <5 months (*n* = 290), infants >5 months (*n* = 711), and mothers (*n* = 146). For (**c** and **d**), a list of outliers not depicted in the plots can be found in Supplementary Data 13. In (**a**−**d**), every dot is a sample, and the dot color indicates age: infant samples (age <5 months) in yellow, infant samples (age > 5 months) in orange, maternal samples in red, and NCs in dark blue. In (**a**−**d**), boxplots visualize the median, hinges (25th and 75th percentiles), and whiskers extending up to 1.5 times the interquartile range from the hinges. Asterisks denote Benjamini−Hochberg-adjusted statistical significance values: *FDR < 0.050, ***FDR < 0.001, ns not significant. 'NA' is used when significance cannot be calculated.

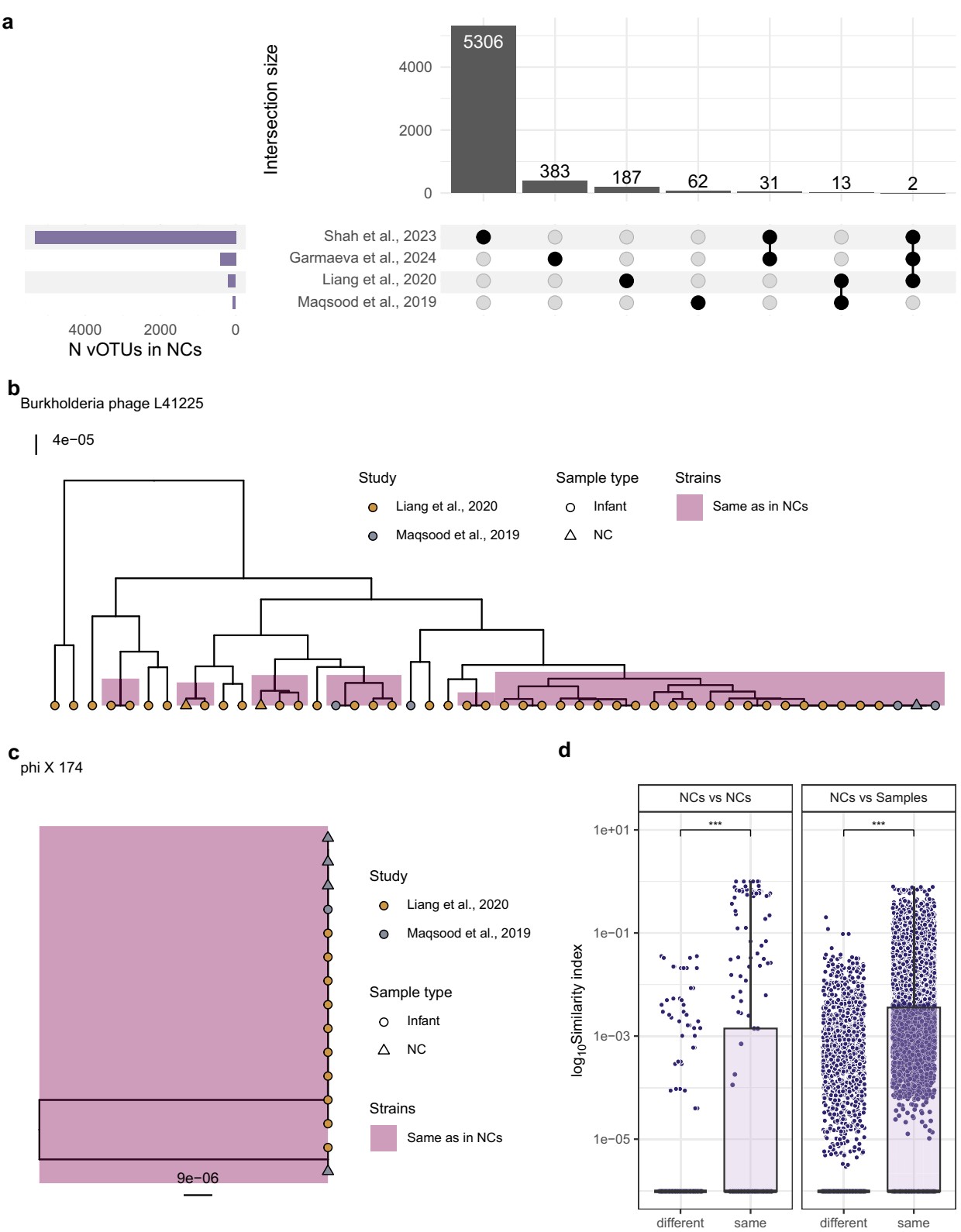

**a**

**b** Burkholderia phage L41225

**c** phi X 174

**d**

any NCs, while some shared up to 100% of their composition at the vOTU level. 71.5% of the samples shared vOTUs with NCs from the same study, with a median of 4.9% (IQR: 1.9–12.9) vOTUs per sample. The median abundance of all vOTUs shared with NCs per sample was 1.7% (IQR: 0.3–8.7). In two of the three studies tested, the abundance of individual NC-shared vOTUs was consistently higher in NCs compared to samples, with 48.3% (N = 28) and 98.9% (N = 2613) of tested NC-

shared vOTUs more abundant in NCs in the Liang et al. and Shah et al. studies, respectively (Supplementary Fig. 8 and Supplementary Data 17). In Maqsood et al., none of the six NC-shared vOTUs we tested were differentially abundant between samples and NCs, and five of these also showed no differences between NCs and samples in the Liang et al. study. For these overlapping vOTUs, *Burkholderia* bacteria were predicted as hosts.

**Fig. 2 | NC-associated contaminants across the four early-life studies.**
**a** Distribution and overlap of vOTUs detected in NCs across studies. The lower-left panel displays the total number of vOTUs identified in NCs per study, while the upset plot on the right illustrates the number of unique and shared vOTUs across studies' NCs. **b, c** Reconstructed ultrametric trees for (**b**) the Burkholderia phage L41225 ('L' followed by a number represents the genome length) and (**c**) phiX174. In (**b, c**), every point is a sample, and the point shape indicates the sample type: infant (circle) and NC (triangle). Point color indicates the study. Cases of identical strain sharing between NCs and samples are highlighted in pink rectangles. **d** vOTU level compositional similarity between NCs (left panel) and NCs vs. samples (right panel) within and across studies. Each dot represents the similarity index between two samples, calculated as 1−Bray−Curtis dissimilarity index (Y-axis is log10-transformed). Data points depicted per category: NCs−NCs (same = 246, different = 840); NCs−Samples (same = 12,485, different = 33,913). In (**d**), boxplots visualize the median, hinges (25th and 75th percentiles), and whiskers extending up to 1.5 times the interquartile range from the hinges. Asterisks denote statistical significance: ***FDR < 0.001.

Given the lower gut viral diversity reported for infants compared to adults[4], we investigated whether contamination levels differed between maternal and infant samples and across infant ages. In the two studies where both infant and maternal samples were available, we found that both the proportion and number of vOTUs shared with NCs from the same study were significantly higher in infants than in mothers (beta = 2.5, FDR = 5.1e-06 in Garmaeva et al.; beta = 11.6, FDR = 2.3e-07 in Maqsood et al.; Supplementary Data 18 and Fig. 3a, b). The estimated contamination decreased with infant age but was significant in only one of the two longitudinal studies (beta = −0.1, FDR = 0.3 in Garmaeva et al.; $beta_2$ = −7.7, FDR = 2e-04 in Liang et al.; Fig. 3c and Supplementary Data 19). These observations suggest that environmental contamination is more likely to be sequenced and detected in early-life gut virome samples due to their low diversity and low overall viral load.

**Strain-level decontamination from NC-shared viruses**
We next assessed whether the overlap identified at the vOTU level extends to strain level, i.e., whether NCs and samples share the same viral strains. We therefore estimated the proportion of samples sharing identical strains with their study's NCs. To do so, we reconstructed virus strains for 5635 vOTUs present in NCs and biological samples (see "Methods"). We found that only 32.6% of these vOTUs ($N$ = 1838) were shared between NCs and at least one biological sample at the strain level, suggesting that the strain detected in the sample might originate from environmental contamination. Furthermore, within each study, 7.7-23.9% of the NC-detected vOTUs were identical to those found in biological samples at strain level.

Across the entire dataset, 71.5% ($N$ = 944) of samples share at least one vOTU with the NCs of their own study, and 85.4% ($N$ = 806, 61.0% of the total number of samples) of those shared at least one strain identical to one detected in NCs. While the median number of strains shared between biological samples and NCs was 2 (IQR: 0−28), the abundance of contaminants exceeded 10% in 11% of infant samples (Fig. 3d, e). Additionally, infant samples shared significantly more strains with NCs than did maternal samples (beta = 3.5, p-value = 1.5e-5, Supplementary Data 20 and Fig. 3f), confirming that early-life gut virome samples are more susceptible to contamination.

In infant samples, a median 34.8% (IQR: 20.0−51.1%) of the vOTUs shared with NCs were identical to the NC strains, whereas the median was 0 (IQR: 0−18.5%) in maternal samples. Therefore, more than half of the vOTUs shared between samples and NCs differed at strain level and likely represented true biological signals. For example, in the Shah et al. study, a Bacteroides phage L6428 from the family *Microviridae* was represented by multiple strains. Although some biological samples shared an identical strain of L6428 with the NCs, more than half of the samples had different strains (Fig. 3g), likely representing Bacteroides phages that are naturally present in the infant gut. We therefore conclude that decontamination of NC-detected viruses from the dataset should be performed at strain rather than vOTU level.

After performing strain-level decontamination, the richness of vOTUs, which represents species-level resolution, dropped by an average 1.5% (IQR: 0.5−5.0) in samples that shared vOTUs with NCs. The number of vOTUs shared with own NCs decreased by 33.3% (IQR: 14.9−50.0) and comprised a median of 2.9% (IQR: 1.0−7.8) of all vOTUs detected per sample. Next, we compared the decontamination results at strain- versus species-level. For the latter, all vOTUs shared with NCs were excluded, leading to a median of 4.9% (IQR: 1.9−12.9) drop in vOTU richness per contaminated sample, which was significantly higher compared to the strain-level decontamination. These results further demonstrate that strain-level decontamination preserves key ecological features of the samples, such as richness.

**Contamination estimation using the NC-associated vOTU catalog**
We next investigated whether biological samples share strains with NCs from other studies, but found that this was rarely the case. Specifically, only 14.5% of the samples ($N$ = 192) shared at least one strain with external NCs. Nonetheless, we did observe moderate concordance in the proportion of strains shared between biological samples and both their own NCs and external NCs (rho = 0.37, p-value = 1.9e-41, Fig. 4a and Supplementary Data 21). However, the sample strains shared with own and external NCs overlapped only by 11.2% on average.

Given the limited amount of strain-sharing with external NCs, we tested if the proportion of vOTUs shared with external NCs could provide an estimate of contamination for studies where NCs are not available and direct decontamination is not possible. Of all the biological samples, 41.1% of samples shared a median of 0.7% (IQR: 0.2−4.8) of vOTUs per sample with NCs from other studies, although this vOTU-sharing was significantly lower than that with own NCs in two out of four studies (Supplementary Data 22 and Supplementary Fig. 9a). The proportion of vOTUs shared with external NCs showed moderate to high correlation to the proportion of strains shared with own NCs in three out of four studies (0.34 ≤ rho ≤ 0.82, Fig. 4b and Supplementary Data 23). We also tested if a proportion of the reads mapped to the genome sequences of vOTU representatives identified in external NCs could be used to estimate contamination, but this provided lower concordance (0.11 ≤ rho ≤ 0.57, Supplementary Fig. 9b and Supplementary Data 24).

These results suggest that sample contamination could be estimated using the proportion of vOTU-sharing between samples and NCs from independent studies, and this estimate could potentially be used as a correction factor in downstream statistical analysis. To facilitate the use of available NCs in quality control of future studies, we have made the database of the negativeome vOTU sequences identified in NCs publicly available[21].

## Discussion
Our results show that viral environmental contaminants detected using NCs affected most of the biological samples in the dataset (61%). Even though samples generally shared a limited number of contaminant viral strains with NCs, the impact of such contamination varied across samples and was greater in infant samples than in maternal ones. This increased susceptibility to contamination in infant virome samples is likely linked to their low viral load[3] and diversity[4] and corresponds with earlier findings linking the contamination impact and sample biomass in other metagenomic samples[5,22,23].

We also showed that biological samples and NCs did not differ in several technical and ecological features, including the number of clean

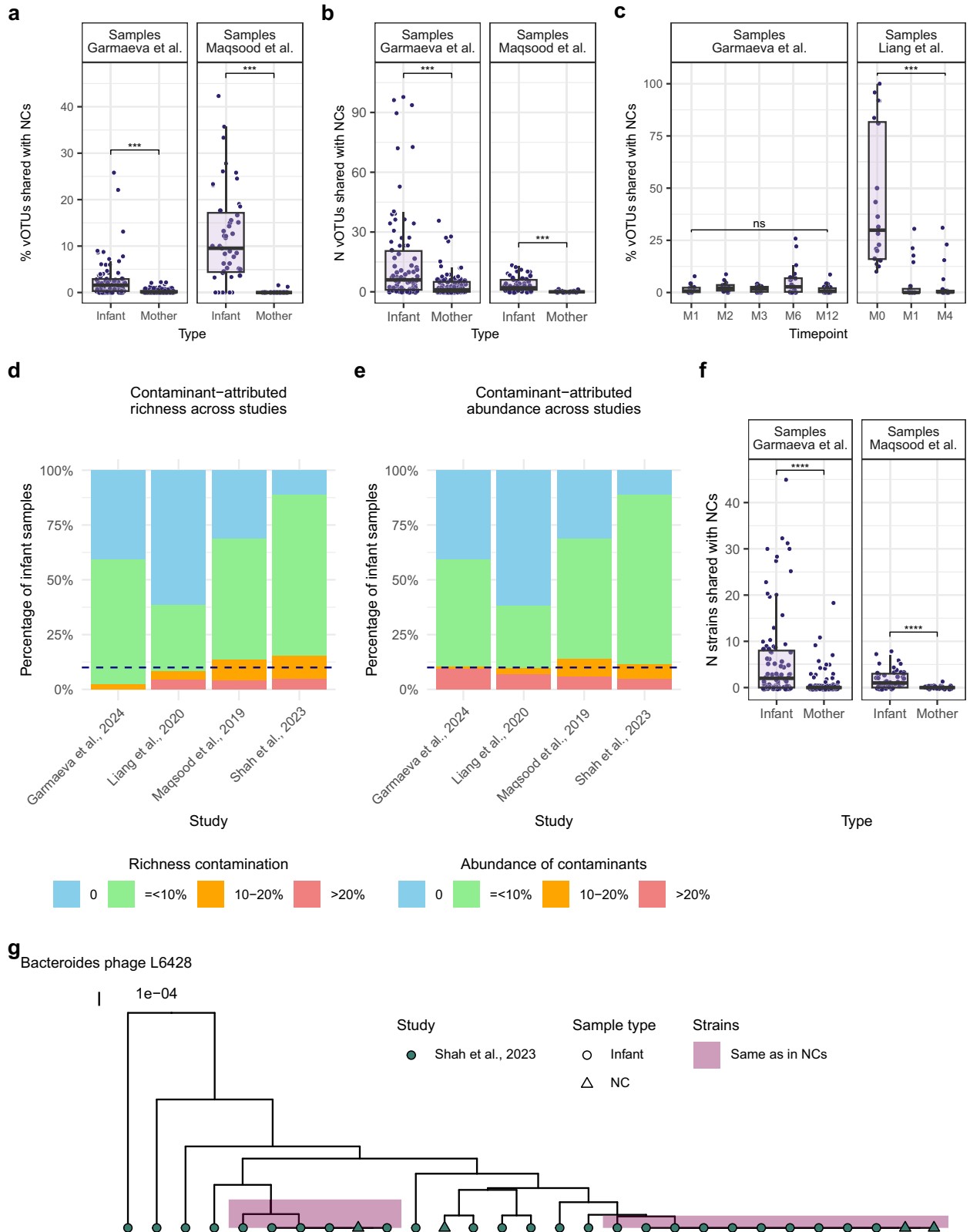

reads and overall diversity. However, features like viral richness and the number of discovered viruses were generally lower in NCs. Despite these differences, infant samples could not be reliably distinguished from NCs using predictive models based on the combined ecological and technical features, echoing findings from bacteriome studies that compared the compositions of NCs and infant meconium samples[24]. This suggests that, while certain metrics capture differences between

NCs and biological samples, they are insufficient for robust classification. Moreover, the significance and direction of feature-based differences often varied across studies, likely reflecting the large variation across studies due to differences in virome extraction, nucleic acid processing, and sequencing methodologies, as well as the overall disparity in the number of NCs and biological samples and age-related variation in the composition and diversity of biological samples.

**Fig. 3 | Estimation of sample contamination based on vOTU and strain sharing with NCs. a** Percentage of sample richness represented by NC-shared vOTUs from the same study in maternal samples compared to infant samples. **b** Number of vOTUs shared with NCs from the same study in maternal samples compared to the infant samples. **c** Percentage of sample richness represented by NC-shared vOTUs per infant timepoint. On the *X*-axis, timepoints are represented by the abbreviations M0, M1, M2, M3, M4, M6, and M12, corresponding to the infant's age in months at the time of sampling (M = month). Data are shown for: Garmaeva et al.: M1 ($n = 11$), M2 ($n = 16$), M3 ($n = 16$), M6 ($n = 20$), M12 ($n = 23$); Liang et al.: M0 ($n = 20$), M1 ($n = 20$), M4 ($n = 20$). **d**, **e** Study-wise distribution of infant samples categorized by the percentage of (**d**). richness and (**e**) abundance of strains shared with NCs. The *Y*-axis shows the percentage of infant samples with different contamination levels, derived and color-coded based on strain sharing with NCs. The

dashed blue line indicates 10% of samples. **f** Number of strains shared with NCs from the same study in maternal samples compared to infant samples.
**g** Reconstructed ultrametric tree for a Bacteroides phage L6428 ('L' followed by a number represents the genome length). Every point is a sample, and the point shape indicates the sample type: infant (circle) and NC (triangle). Point color indicates the study. Cases of identical strain sharing between NCs and samples are highlighted in pink rectangles. In (**a**–**c**, **f**, **g**), each dot represents a sample. Boxplots visualize the median, hinges (25th and 75th percentiles), and whiskers extending up to 1.5 times the interquartile range from the hinges. Asterisks denote statistical significance: ***FDR < 0.001, ****FDR < 0.0001, ns not significant. In (**a**, **b**, **f**) data are shown for Garmaeva et al.: maternal ($n = 119$), infant ($n = 86$); Maqsood et al.: maternal ($n = 27$), infant ($n = 51$).

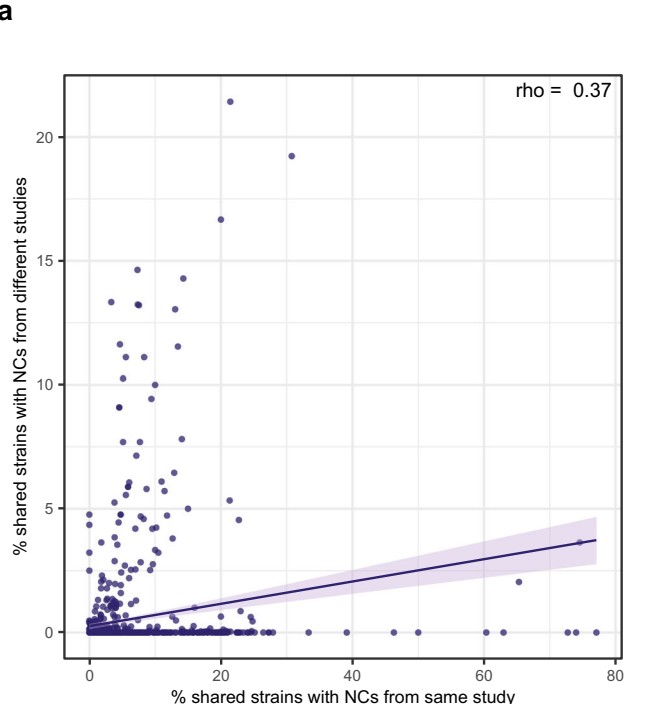

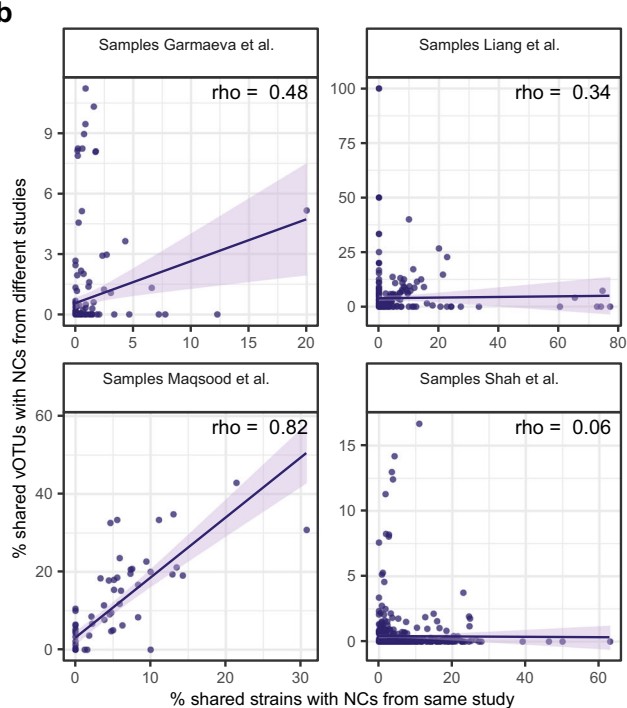

**Fig. 4 | Contamination levels estimation using NC-detected sequences from external studies. a** Correlation of the percentage of strains shared between samples and internal versus external NCs. **b** Correlation of the percentage of strains shared between samples and internal NCs with the percentage of vOTUs shared between samples and NCs from external studies. In (**a**, **b**), each dot is a sample. Data are shown for 1254 samples. Per study distribution: Garmaeva et al. ($n = 205$), Liang

et al. ($n = 324$), Maqsood et al. ($n = 78$), and Shah et al. ($n = 647$). The percentage of strains and vOTUs shared with NCs is calculated per sample as the number of strains or vOTUs shared with NCs divided by the sample richness. The solid line represents the fitted linear regression, and the shaded band denotes the 95% confidence interval of the model. Spearman correlation rho's are depicted in the upper right corner of each panel.

Although the contamination was largely study-specific, we did identify a few previously reported contaminants[5,6,20], but with rather low overlap. The viral sequences identified in NCs in our study thus extend existing catalogs of lab-associated environmental contaminants, and we have made them publicly available to aid other researchers in assessing contamination[21]. It should be noted that 89.2% of the vOTUs detected in NCs originated from a single study by Shah et al., skewing the catalog toward that dataset. This study also had the highest total number of raw and clean reads per sample and the highest number of vOTUs detected, with 5% of them found in NCs, compared to 0.4–2.8% in the other studies. This slightly elevated proportion may be explained by deeper sequencing and by differences in VLP extraction protocols between studies. The latter could be influenced by the pore size of the membrane filter used for bacterial and large particle removal, the number of filtration steps used, the methods used for VLP concentration, variations in nuclease treatment and DNA extraction kits across the four studies, and the use of the multiple displacement amplification (MDA), which is known to

increase sequence sensitivity by amplifying both biological signals and contaminants[5,25].

We also identified that using strain-level rather than vOTU-level decontamination with internal study NCs helps preserve key ecological metrics such as sample richness. We therefore recommend that decontamination be performed at the strain level because this provides higher precision for removing the exact sequences of contaminating viruses and preserves true biological signals. We have also developed a pipeline for strain-level decontamination[26] (https://github.com/GRONINGEN-MICROBIOME-CENTRE/NCP_VLP_project/).

We acknowledge the existence of scenarios when proper NCs are not available, such as the reuse of previously generated datasets. In this study, we have shown that strain-level decontamination using sequences from external NCs has a lower concordance and efficiency than using internal NCs. However, the contamination level estimated using the proportion of vOTUs shared with external NCs correlates with the proportion of contaminant strains shared with internal NCs. Therefore, vOTU-sharing with the negativeome vOTU catalog from

this study provides a rough estimate of the general sample contamination level. While this method cannot replace direct decontamination with internal NCs, the estimate of contamination levels it offers can support sample quality control and the inclusion of contamination as a correction factor in statistical analyses.

Overall, our study shows best practices in decontamination using internal NCs, provides a database of sequences detected in the NCs from publicly available gut virome studies for future reuse, and identifies scenarios in virome studies where this resource is critical for assessing and addressing contamination. Given our results regarding the higher susceptibility of early-life virome samples to contamination, we emphasize that NCs from multiple sources and processing stages must be included in future virome studies. We anticipate that public data-sharing of NCs from such studies will enhance the NC database and improve the quality of early-life gut virome data for future reuse.

## Methods

### Publicly accessible datasets of VLP-enriched early-life human gut samples

To explore the impact of contamination on early-life gut virome studies, we selected studies that: (1) were focused on the early-life infant gut virome, (2) employed VLP sequencing, and (3) had publicly available sequencing data, including NC sequencing data, as of January 2024. Detailed descriptions and metadata of the studies mentioned below can be found in the original articles[2–4,13]. Here, we briefly summarize the set-up of each study.

Garmaeva et al.[4] was based on samples collected longitudinally from 32 mother–infant pairs of the Dutch LLNEXT cohort[27]. In total, 86 VLP samples were recovered from 32 infants (including two twin pairs) during the first year of life. One hundred nineteen maternal samples from 30 mothers were collected longitudinally from 28 weeks of pregnancy to 3 months postpartum. Four negative buffer controls were isolated with the samples: three failed sequencing and the fourth is included in the current study.

Liang et al.[3] was carried out in Pennsylvania Hospital, Philadelphia, USA, and includes VLP data for 185 samples collected longitudinally from 144 infants from 0 to 4 months and 21 samples collected from 21 children from 2 to 5 years. For all infant samples, both DNA and RNA VLP sequencing was done. For samples of children from 2 to 5 years, only DNA VLP sequencing was done. We included both the DNA and RNA sequencing samples in our study and analyzed them as separate samples. The Liang et al. study included 19 NCs of different origins, including empty diaper samples, empty stool container samples, and reagent-only samples. For each NC, both DNA and RNA sequencing were performed, further doubling the number of NC samples to 38. While all these NCs were initially included in our study, we did not discover any putative viral sequences in 12 out of 38 NC and did not detect any vOTU in 18 out of 38 NC samples.

Maqsood et al.[2] is a study conducted in St. Louis, Missouri, USA that analyzed the birth stool of twins alongside samples collected from their mothers. Samples from 51 infants and 27 mothers were available for this study. Two different types of NCs were included: buffer NCs ($N = 4$) isolated to describe general contamination and NCs with added Nematoda virus (Orsay NC, $N = 4$) to assess levels of cross-contamination. Both types of NCs were isolated along with the samples and were analyzed in our study.

Shah et al.[13] is a study done in Copenhagen, Denmark that used samples collected as part of the COPSAC2010 cohort[28]. Samples were collected from 647 one-year-old infants cross-sectionally and isolated along with 8 buffer NCs.

### Sequencing reads: quality control and assembly

We carried out read quality control and assembly, adjusting for study specifications such as the presence of MDA during the VLP library construction.

**Garmaeva et al.** Raw reads underwent adapter trimming with the bbduk.sh script from BBTools[29] (v39.01) using the following flags: ktrim = r k = 23 min = 11 hdist = 1 tpe tbo. Human read removal and read quality trimming and filtering were performed using kneaddata[30] (v0.10.0) and human reference genome (GRCh38p13), followed by quality trimming with the option –trimmomatic-options "LEADING:20 TRAILING:20 SLIDINGWINDOW:4:20 MINLEN:50". Quality-trimmed and filtered reads were assembled using SPAdes[31] (v3.15.3) with "–meta" mode. For the one NC sample, SPAdes failed to perform internal sequencing read processing. We therefore performed read error correction with tadpole.sh (parameters: mode = correct, ecc = t, prefilter = 2) from BBTools[29] (v39.01). Error-corrected and deduplicated reads of NC were assembled with SPAdes[31] (v3.15.3) using the "–meta" and "–only-assembler" modes.

**Maqsood et al. and Liang et al.** First, we excluded samples SRR8653201, SRR8800143, and SRR8800149 from the Liang et al. study, following the original article authors' recommendations. Raw read quality control and filtering were then performed as described above. Next, to maximize MDA-treated samples' de novo assembly performance, we performed read deduplication and assembly in the uneven coverage-aware mode[32]. Briefly, read error correction was done with tadpole.sh, as described above, and read deduplication was performed using clumpify.sh (dedupe = t, subs = 0) from BBTools[29] (v39.01) to remove identical sequences. Read assembly was done with SPAdes (v3.15.3) using "–sc" mode[33,34].

**Shah et al.** No quality control of the sequencing reads was performed since the deposited samples were already filtered for low-quality and human reads and deduplicated. Assembly was performed with SPAdes (v3.15.3), using "–sc" mode[33,34].

### Putative virus sequence prediction from the metaviromes

Putative virus sequences were predicted per sample using assembled contigs >1000 bp. To maximize the likelihood of virus sequence recognition, we applied four different tools: (1) VirSorter2[35] (v2.2.4) with the flag "–include-groups "dsDNAphage,RNA,NCLDV,ssDNA,lavidaviridae"", (2) DeepVirFinder[36] (v1.0) with a threshold for the score of ≥0.94[37], (3) geNomad[38] (v1.7.4) in "–end-to-end" mode with enabled score calibration, and (4) VIBRANT[39] (v1.2.1) with the open reading frames predicted using prodigal-gv[38,40] (v2.9.0-gv) as an input. All contigs identified as viral by at least one of these tools were selected for further analysis.

We also attempted to extend identified putative virus sequences using COBRA[41] (v1.2.3) to recover more complete viruses. To trim host-associated regions from provirus sequences, we subsequently employed geNomad[38] (v1.7.4) and CheckV[42] (v1.0.1). Sequences containing direct or inverted terminal repeats identified by geNomad were exempted from running through CheckV. Genome completeness was estimated for all sequences after prophage-pruning using CheckV.

### vOTU processing and RPKM table creation

971,583 viral contigs were further dereplicated using the Minimum Information about an Uncultivated Virus Genome recommended cut-offs for the species-level rank: 95% average nucleotide identity (ANI) over 85% alignment fraction (relative to the shorter sequence)[43], resulting in 307,938 vOTU representatives. To estimate vOTU abundances, we used Bowtie2[44] (v2.4.5) in 'end-to-end' mode to align reads to the vOTU-representative genomes, followed by count table generation using SAMTools[45] (v1.18) and BEDTools[46] (v2.30.0). Read counts with a breadth of coverage less than $1 \times 75\%$ of a contig length were set to 0 to remove spurious Bowtie2 alignments[47]. Final read counts were transformed to reads per kilobase per million reads mapped (RPKM) values, which were subsequently used for downstream analyses. We further removed vOTU representatives from the

RPKM table if they were shorter than 1000 bp or had fewer viral genes than host genes based on CheckV assignment. Sequences identified as plasmids by geNomad[38] (v1.7.4) were also removed from the table. The final RPKM table included 193,970 vOTU representatives detected in 1368 samples, including 55 NCs.

## vOTU taxonomic profiling and host prediction

We used geNomad[38] (v1.7.4) taxonomic assignment for the resulting vOTU representatives. Using the taxonomic assignment, we derived information about viral nucleic acid type and predicted host domain (eukaryote or prokaryote) from ICTV[48]. The iPhoP[49] (v1.3.3) framework with database "Aug_2023_pub_rw" was used for host prediction for vOTU representatives. For this analysis, we excluded vOTU representatives identified as eukaryotic viruses via taxonomy assignment. The genus-level host prediction with the top confidence score per vOTU representative of at least 50% genome completeness, as predicted by CheckV, was selected for further analysis. We carried out species-level host prediction for two phages predicted to infect *Burkholderia* using a combination of the iPhoP host genome assignment and blastn against nt database (update from 2024/08/14) with BLAST +[50] (v2.13.0), as BACPHLIP[51] (v0.9.6) predicted one of the phages to be temperate. Based on the identified ANI of >96% and 76% query coverage, the host was narrowed down to *B. cepacia* complex species.

Sequences identified in the NCs were queried against the NT database (updated 25 November 2023) using blastn search with an *e*-value of $1 \times 10^{-3}$. The best hit was selected based on *e*-value, query coverage, percent identity, and alignment length. These hits were then compared with the LCA viral sequences reported by Asplund et al.[6] to identify overlapping hits. Hits with query coverage <30% were discarded as potentially spurious matches. NT-hits to viral taxa that did not match LCA are reported in Supplementary Data 15.

## Virus strain reconstruction and comparison

For the 5635 vOTU representatives present in the biological samples and at least one NC, we reconstructed consensus sequences using inStrain[52] (v1.9.0). Briefly, the sorted sequence alignment map files of samples and NCs where the vOTU representative of interest was identified were processed with the 'profile' module of inStrain in the "–database_mode" with a minimum coverage requirement for variant calling of 1 ("–min_cov 1"). Next, we used the 'compare' module of inStrain to estimate the pairwise genome similarity among all reconstructed consensus sequences. Only regions with a minimum of 1× coverage were included in the analysis. Consensus sequence pairs with less than 75% of the genome available for comparison were excluded from the analysis. Population-level ANI (popANI) values were used to compare the similarity between strains belonging to the same vOTU. Virus strains were considered to be shared between samples if their compared genomic regions exhibited ≥ 99.999% popANI[52].

## Virus sequence decontamination from NC sequences

To remove virus sequences shared between biological samples and NCs at strain level, we zeroed out the RPKM values of the relevant vOTU representatives. To account for potential limitations due to varying sequencing depth between samples and NCs, we considered all cases of strain-sharing, including those with less than 75% of the genome available for comparison. For the comparison of strain- and species-level decontamination, vOTUs detected in the NCs and samples from the same study were zeroed out.

## Contamination estimation using sequences from the NC-associated vOTU catalog

To assess the feasibility of using the negativeome vOTU-based catalog[21] to estimate contamination in the absence of internal NCs, we compared two metrics: (1) the percentage of reads from the biological samples that mapped to the catalog and (2) the percentage of vOTUs shared with the catalog per biological sample. We evaluated the reliability of these metrics by assessing their concordance with the percentage of sample richness shared with the studies' internal NCs at strain level. In the first approach, for each study, we created a sub-database of vOTUs detected in the NCs of the other three studies and mapped reads from biological samples to these vOTUs using Bowtie2[44] (v.2.5.1) with "–very-sensitive" flag. The contamination metric was calculated as the proportion of reads mapped to contigs with more than 50% or 75% breadth coverage. This approach showed low to moderate concordance with the percentage of sample richness shared with studies' internal NCs at strain level (0.11 ≤ rho ≤ 0.57, Spearman correlation), likely due to spurious mapping or artificially inflated mapping coverage due to multimapping artifacts. In the second approach, we calculated the percentage of vOTUs shared per biological sample with the catalog using the vOTU table. Specifically, for each study, we determined the percentage of richness in a biological sample that overlapped with NCs from the other three studies. This approach demonstrated moderate to high concordance with the percentage of sample richness shared with studies' internal NCs at the strain level for three out of four studies (0.34 ≤ rho ≤ 0.82, Spearman correlation, Fig. 4b).

## Data visualization

Results were visualized in graphical form using a set of custom R scripts (R v4.2.3), including calls to functions from the packages ggplot2[53] (v3.5.0), tidyverse[54] (v. 2.0.0), ggrepel[55] (v.0.9.6), ggforce[56] (v.0.4.2), and patchwork[57] (v.1.3.0). All boxplots were prepared using ggplot2 and represent standard Tukey type with IQR (box), median (bar) and Q1−1.5 × IQR/Q3 + 1.5 × IQR (whiskers). Phylogenetic trees were built based on the estimated pairwise genome dissimilarity (1−popANI), using only consensus sequence pairs for which at least 75% of the genome was available for comparison. Hierarchical clustering was applied to the matrices of genome dissimilarity using the function hclust() from the R package stats[58] v.4.2.1. Clustering trees were then converted into phylogenetic trees with the function as.phylo() from R package ape[59] v.5.7.−1. Phylogenetic trees were visualized using the ggtree() function from the R package ggtree[60] (v 3.14.0). The results for the differential abundance of the vOTUs between NCs and biological samples were visualized using the R package EnhancedVolcano[61] (v1.24.0).

## Statistics and reproducibility

The current study uses biological samples and NC samples from publicly available datasets. No prior selection was applied to the samples, and no statistical method was used to predetermine sample size. Of the 1321 biological samples and 55 NCs retrieved, we used 1313 and 55, respectively, for the statistical analyses.

Statistical analyses were performed using R[62] (v4.2.3). For detailed information regarding the tests used to assess the significance of results, see Supplementary Data. To estimate differences between samples (N = 1313) and NCs (N = 55) in features such as number of clean reads, reconstructed contigs, discovered viral sequences, viral richness, and viral diversity, we used linear mixed models (lme4[63], v1.1-23, and lmerTest[64], v3.1-3) (Supplementary Data 2–4, 6–7). In each comparison, the studies were analyzed both together and separately. When analyzed together, the model was corrected for the type of nucleic acid used as the template for viral sequencing ('RNA' vs. 'DNA'). The study and subject group were included as nested random factors. For the biological samples, the subject group was defined as the subject ID to account for the repeated measurements, while NCs were grouped by the source within each study (Supplementary Data 1). When analyzed separately, the subject group was used as a random factor. The model used for Liang et al. was also corrected for the nucleic acid type of the sample. The study by Garmaeva et al. was excluded from the per-study

analysis due to the availability of only one NC. All data for the features were subjected to inverse-rank transformation prior to analysis. We then used the same approach to estimate differences in the proportion of reconstructed viruses with at least 50% completeness to the total number of all reconstructed viruses per sample between the samples ($N = 1254$) and NCs ($N = 37$), taking into account only viruses with at least 50% completeness (Supplementary Data 5).

To assess whether genomic and ecological features can distinguish NCs from biological samples based on their hypothetical sampling age, we used data from the longitudinal studies by Garmaeva et al. and Liang et al. We constructed a linear model with age (in months) as the outcome variable, using only infant biological samples. For the Liang et al. dataset, we only included samples collected at months 0, 1, 3, and 4. The predictors included the number of reconstructed contigs, discovered viral sequences, genome completeness, and viral richness. All predictor features were inverse-rank transformed prior to model fitting.

For composition difference analysis, only vOTUs represented by viruses with at least 50% genome completeness were considered (Supplementary Data 9 and 11). Significance was assessed using linear mixed-effects models as described above. Proportional compositional features were log-transformed prior to analysis. A linear model, rather than linear mixed-effects models, was used for the studies by Maqsood et al. and Shah et al. P-values were adjusted using the Benjamini–Hochberg correction method. Viral diversity (Shannon index) was calculated using the *diversity()* function in the vegan[65] package v2.6-4.

For the viral composition analysis, the proportions of vOTUs with assigned nucleic acid type and predicted hosts were calculated using only vOTU representatives with at least 50% completeness, based on a binary RPKM table (presence/absence data). These data were log-transformed prior to calculations (Supplementary Data 9 and 11).

Beta diversity analysis was performed at the vOTU and host-based vOTU aggregate levels using Bray–Curtis dissimilarity. The Bray–Curtis dissimilarity between samples was calculated using the function *vegdist()* from R package vegan[65] (v2.6-4). Visualization of sample beta diversity was performed with non-metric multidimensional scaling using the function *metaMDS()* from R package vegan[65] (v2.6-4) with two dimensions. To assess the similarity between NCs and samples, we subtracted Bray–Curtis dissimilarity indices from 1. A two-sided Wilcoxon rank sum test, conducted through a permutation approach with $N = 10{,}000$ iterations, was performed to determine the significance of differences observed between groups (Supplementary Data 16).

The percentage of vOTUs shared between biological samples and NCs was calculated as the proportion of shared vOTUs to the total number of viruses detected per sample. A two-sided Wilcoxon rank sum test, conducted through a permutation approach with $N = 10{,}000$ iterations, was performed to determine the significance of observed differences in sample vOTU-sharing with NCs from the same study versus NCs from different studies (Supplementary Data 22). To assess the correlation between sample strain-sharing with same-study NCs versus sample vOTU and strain-sharing with different studies' NCs, we calculated a study-wise Spearman correlation coefficient (Supplementary Data 21, 23, and 24). P-values were adjusted using the Benjamini–Hochberg correction method.

For the analysis relating sample vOTU-sharing with the same-study NC to participant type (infant versus mother), we used only biological samples from Garmaeva et al. ($N = 205$) and Maqsood et al. ($N = 78$), as maternal samples were only available in these studies (Supplementary Data 18). For Garmaeva et al., significance was estimated using linear mixed-effects models, with the subject group added as a random factor. For Maqsood et al., we used a linear model because only one timepoint was available per participant. P-values were corrected using the Benjamini–Hochberg method.

For the analysis of vOTU-sharing with same-study NCs over time, we used only the two studies with multiple timepoints per subject (Supplementary Data 19): Garmaeva et al. ($N = 205$) and Liang et al. In Liang et al., only one cohort had multiple timepoints per infant available, so only this cohort's samples ($N = 146$) were used for the analysis. The RNA samples from Liang et al. were excluded from this analysis. Significance was assessed separately for each study using linear mixed-effects models, with the subject group included as a random factor. P-values were corrected using the Benjamini–Hochberg method.

For the identification of vOTUs that were differentially abundant between NCs and biological samples, we included only the vOTUs detected in at least two samples and two NCs to ensure robust comparisons. This filtering reduced the number of vOTUs analyzed and led to the exclusion of the Garmaeva et al. study, which had only one NC. The final dataset comprised 6 vOTUs from Maqsood et al., 58 vOTUs from Liang et al., and 2914 vOTUs from Shah et al. The analysis was done separately for each study using linear mixed models (lme4[63] and lmerTest[64]) with the subject group included as a random factor (Supplementary Data 17) to assess significance and using log2 fold change to evaluate differences in abundance.

## Data availability

All data used in this study are publicly available, with sequencing data for Maqsood et al., Liang et al., and Shah et al. accessible via the European Nucleotide Archive (project numbers PRJEB33578, PRJNA524703, and PRJEB46943, respectively). For Garmaeva et al., sequencing data are available in the European Genome-Phenome Archive (EGA) repository (study ID: EGAS00001005969). The source and processed data generated in this study, including redundant virus sequences and vOTU representatives, along with their metadata, and the database of virus sequences identified in the negative controls (v1.0.0), have been deposited in the FigShare repository under https://doi.org/10.6084/m9.figshare.27170739.v2. Other datasets or databases used in the present study are: Human reference genome GRCh38.p13 [https://www.ncbi.nlm.nih.gov/datasets/genome/GCF_000001405.39/] and iPHoP database (Aug_2023_pub_rw) [https://portal.nersc.gov/cfs/m342/iphop/db/iPHoP.latest_rw.tar.gz].

## Code availability

The code used in this study can be found at: https://github.com/GRONINGEN-MICROBIOME-CENTRE/NCP_VLP_project and https://doi.org/10.5281/zenodo.15682695[26].

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

## Acknowledgements

We thank Kate Mc Intyre for editing the manuscript. We also thank the Genomics Coordination Center and the Center for Information Technology of the University of Groningen for their support and for providing access to the Gearshift and Hábrók high-performance computing clusters. S.G. was supported by a scholarship from the Graduate School of Medical Sciences, University of Groningen. S.G. was awarded a de Cock-Hadders Stichting grant (2021-08). Furthermore, this project was funded by the Netherlands Organisation for Scientific Research (NWO): NWO Gravitation Exposome-NL (024.004.017) to A.K. and A.Z. and an NWO-VICI VI.C.232.074 to A.Z. A.Z. was also supported by the EU Horizon Europe Program grant INITIALISE (101094099). A.K. was supported by ZonMW ME/CFS grant 10091012110017.

## Author contributions

S.G. designed the study. N.K. gathered and prepared the data. N.K., A.K., and S.G. analyzed the data. N.K., A.Z., and S.G. wrote the manuscript. A.K. provided advice for statistical methods. N.K., A.K., A.Z., and S.G. critically reviewed the manuscript.

## Competing interests

The authors declare no competing interests.
