## [Transparent Peer Review file · Nature Communications]

Negativeome characterization and decontamination in early-life virome studies

Corresponding Author: Dr Sanzhima Garmaeva

Version 0:

Reviewer comments:

Reviewer #1

(Remarks to the Author)

Kuzub and colleagues used 1321 viral metagenomes of infant gut and 55 negative controls (NCs) from four public studies to address the viral contamination that are potentially present in the biological samples. The viral analysis pipeline is standard and accepted currently, and they have characterized the viral composition from biological samples and NCs, and then made comparisons at species and strain level. The main benefit of the current study is to provide the database of vOTU sequences identified in NCs that is publicly available for other studies. Beyond that, I personally think that the novelty of this study is limited and makes little advances to the field of early life gut microbiome, and some conclusions are too arbitrary.

Major comments:

Line 56-58: the author concluded that NCs did not differ from biological samples in any major genomic and ecological features, including the richness of viral sequences or vOTUs. However, we can see ten times differences in the number (64-594 vs. 0-33). I think this is largely due to the viral variations among biological samples from infants from birth to five year. It is known that the viral richness increases significantly in early life, although the linear mixed-effects model was used for the statistical analyses in order to account the repeated measurement. The quite low number of NCs may also lead to the non-significant results. Therefore, how/what criteria did the author choose these four studies?

Line 76-77: the author concluded that the viruses identified in NCs resemble the viruses of the human gut, did the author calculate the beta diversity between NCs and biological samples? Based on figure 1e, there are more viral taxa in biological samples than that of NCs.

Line 110-113: This is interesting finding and may be useful for studies without setting NCs. However, I think setting own NCs in study is more important, and with the reduce in sequencing cost, setting a few NCs will be easy and valid for own findings.

Line 124-126: the proportion of vOTUs shared with NCs was significantly higher in infants than in mothers. As we know, the richness of viruses in infants is much lower than that of adults, which may lead to higher proportion of viral contamination. Thus, it is hard to draw that conclusion that lab-induced contamination is more likely to be sequenced and detected in early-life gut virome samples.

Line 133-135: may I ask how figure 2f reflect the finding here?

Line 156-164: the overall conclusion is quite general and these are the practical tips for sequencing, such as contaminations may come from multiple sources at different processing stages, and setting controls are very necessary. Additionally, "infant feces sampled closer to birth are more prone to contamination and share more vOTUs with NCs than feces sampled later in life." is needed to reanalyzed as commented above.

Minor comments:

Line 136: how is the 71.5% obtained? Is it not based on 806/1321?

Line 94-95: Figure 2c is hard to understand.

The meaning of asterisk should be provided in all legends.

Figure 2f: what is meaning with "shah_kid122_N774_L6428_K6.1_E0_P0_F0"?

(Remarks on code availability)
The code is clear.

Reviewer #2

(Remarks to the Author)

In this study, Kuzub et al. investigate the impact of environmental contamination on virome datasets from human gut studies. Using publicly available virome data from infant cohorts, the authors compared biological samples with negative controls (NCs) to identify vOTUs that were shared between the two types of samples. Such sharing events would then imply possible contamination. They found that while most biological samples indeed shared at least one vOTU with the NCs, these viruses most commonly differed at the strain-level, making them less likely to reflect an actual contamination event. The authors thus recommend performing virome decontamination at the strain level rather than broader taxonomic levels, and also show that even when NCs are unavailable, decontamination against NCs from other studies is nonetheless useful.

Overall, I found the study to be interesting and valuable, emphasizing the importance of estimating and managing contamination in virome datasets. However, I found the authors' results and conclusions to be somewhat inconsistent, and the impact of contamination appears overstated. For example, the authors claim that 71.5% of samples contain at least one contaminant. But, after decontamination, the authors show this only reduces 1.5% of overall richness, which suggests these contaminants represent a very small portion of the sample diversity. It is important to include additional analyses to better understand the impact of the contamination identified on any subsequent biological conclusions. I provide below more specific comments:

1. Though the various analyses at vOTU- and at strain-level are well-described, the bottom-line recommendations for other researchers when performing decontamination should be sharpened and justified. Specifically, given that one of the main messages of the study is that decontamination should be done at the strain-level rather than the vOTU level, it is unclear why the authors nonetheless offer readers an NC-associated vOTU catalog to estimate contamination (lines 110-113), and conclude the main text with a recommendation to use vOTUs (156-160). If this is due to technical considerations (e.g., hurdles of strain-level analysis), and vOTU sharing is a good-enough proxy, this argument should be made in the paper.
2. I agree with the premise that strain-level analysis does provide more accurate decontamination procedures. However, the authors need to expand their analyses to better support how much of an impact this actually makes. For example, how does their decontamination procedure change the results of the original publications cited, if at all? As mentioned above, if removing contamination only filters out 1.5% of the sample diversity, then I am not sure how biologically meaningful this whole procedure is in practice.
3. The results are presented in a way that appears to overstate the extent of contamination discovered. Upon closer inspection, the differences are much more subtle. For instance, the authors use statistics such as number of vOTUs, contigs, etc. to argue that biological samples and NCs are indistinguishable. However, in Fig. 2c, where the sample composition is taken into account, the median similarity observed between samples and NCs appears to be close to 0 for all the comparisons. This implies that at a compositional level, biological samples and NCs are in fact very distinct. The authors need to be more upfront about this. I suggest performing a PCoA analysis per study and include a figure showing how biological samples and NCs cluster in a 2D space.
4. Related with the above point, most of the results are presented in boxplots that are log-scaled (even after relative abundance normalization). This gives a misleading impression that the distribution values are much higher. I suggest the authors to not use log-transformation at least when presenting differences in terms of proportions (e.g., Fig. 2e and Supplementary Fig. 4b).
5. Furthermore, beyond showing what proportion of vOTUs are shared, it is important to also accurately assess which ones are not differentially abundant between NCs and biological samples using proper statistical tests. The authors should perform a statistical analysis (e.g., using mixed effects models) to identify which vOTUs are or are not statistically different between groups. The results could then be illustrated in a heatmap to present the abundance variation of selected vOTUs within each sample group.
6. The authors should clarify in the Abstract what proportion of total vOTUs detected were contaminants (I assume this is only 3%, 5,984/193,970) and how much of the abundance of each sample on average was composed of contaminant OTUs. In addition, including a figure (e.g., Venn diagram) showing what proportion of vOTUs were found to be unique to biological samples, controls or shared would be very helpful.
7. In the main text when first mentioning vOTU creation, it is worth defining that vOTUs roughly reflect species-level clusters, to make it clear that when moving to strain-level you are increasing rather than decreasing resolution.
8. The human gut is not considered a low-biomass environment per se. The issue of contamination in the samples here studied relate with the viral enrichment protocols. Important to clarify this.
9. Some sentences should be rephrased to improve readability and clarity. Examples: lines 92-94, 102-104, 104-106, ...
10. Throughout the manuscript, when noting a p-value, FDR, or another statistic, it should be clear what statistical test was used (e.g., line 48, line 51, etc.).

11. Line 77-78: Improve conciseness - "at least at the highest taxonomy and predicted host levels"?
12. In Supplementary Figure 1 the term "vOTU" is used but not defined anywhere. A typo maybe?
13. Across several figures, 'NA' is listed instead of significance levels / "ns" when comparing boxplots. If this is on purpose it should be noted in the legend.
14. Line 155: When you note that your pipeline has been made available, please refer to where readers can find it.
15. The term "compositional similarity", used in several places in the paper, is slightly ambiguous and can refer to a few different things. Consider replacing it with a more concise term.
16. The methodology of defining which contigs are viral based on the 4 different tools, as described in lines 232-234, is unclear. Did you mean you chose only contigs that had been labeled as viral by all methods?
17. Line 243: The correct term should be "clustering" rather than "dereplication".
18. To place the current work in context of other similar studies, it may be interesting to compare the current catalog of contaminating viral sequences to other catalogs published (using a different approach) such as this one: [10.1016/j.cmi.2019.04.028](https://doi.org/10.1016/j.cmi.2019.04.028). Alternatively, you may note other approaches for viral decontamination.
19. A few highly relevant papers seem to be unmentioned and should be referred to in the introduction: [10.3389/fmicb.2021.745076](https://doi.org/10.3389/fmicb.2021.745076), [10.1016/j.cmi.2019.04.028](https://doi.org/10.1016/j.cmi.2019.04.028).

(Remarks on code availability)

I reviewed the above GitHub repository which appears to contain all the relevant code with sufficient documentation. However, I have not run the code myself.

Reviewer #3

(Remarks to the Author)

(Remarks on code availability)

Version 1:

Reviewer comments:

Reviewer #1

(Remarks to the Author)

The authors have done additional analyses to improve the confidence of the findings. However, the limited novelty and workload of this study are still held. Lots of the findings or conclusions have already been acknowledged or can be predicted.

Regarding the conclusion that NCs did not differ from biological samples in any major genomic and ecological features, including the richness of viral sequences or vOTUs, it is predicted that this non-significance is due to the viral variations among combined biological samples from infants covering the first five year of life and adults (mothers). The very low number of NCs in each study or study-specific variations in their viral composition also make the statistics non-significant.

Secondly, the author found that early-life VLP samples share more vOTUs with NCs and are therefore more susceptible to contamination. Are there any biological reasons behind this phenomenon? If it is simply caused by the low diversity of virome in early life, this is easy to understand and has been recognized for all low-biomass samples, and thereby the findings here may make no advances to our current understanding.

Finally, contamination has been frequently discussed and concerned during sample preparation and sequencing. Therefore, it is still recommended to set own NCs in each study, and with the reduce in sequencing cost, setting a few NCs will be easy and valid for own findings, instead of downloading these deposited sequences and blasting reads to them, which makes the situation more complicated. Besides, two studies (out of four studies in total) show very low correlations when calculating the proportion of vOTUs shared with external or own NCs.

Line 215-218: why the number of vOTUs shared with own NCs decreased by 33.3%, not 100% after performing strain-level decontamination?

(Remarks on code availability)

I have reviewed the code in GitHub repository. However, I did not run the code by myself.

Reviewer #2

(Remarks to the Author)

I have reviewed the revised manuscript and the authors' detailed responses to my comments. In general, I am pleased with the improvements of the manuscript and the thorough revisions made. I have only two remaining comments/concerns:

1) It would be better to show the new Venn diagram provided on a per study level (Supplementary Figure 7). As the majority of negative control (NC) vOTUs were obtained from the dataset of Shah et al. (5,306 out of the total 5,984 vOTUs) it would be beneficial to see the intersection for each study separately.

2) Related with the above point, authors should acknowledge that the majority of their NC vOTU catalog is coming from just 8 NCs from Shah et al. This should also be more thoroughly discussed as one of the limitations of the study. It would also be important to provide more details on how the samples were originally processed in this study as part of the Methods section. Is there a particular reason why this one study has such high levels of contaminant vOTUs? I could not find any information about these controls in the original paper (<https://doi.org/10.1038/s41564-023-01345-7>).

(Remarks on code availability)

Reviewer #3

(Remarks to the Author)

(Remarks on code availability)

Version 2:

Reviewer comments:

Reviewer #1

(Remarks to the Author)

The authors have addressed all my concerns, and the limitation has also been discussed in the revised manuscript. I have no further comments.

(Remarks on code availability)

I have reviewed the code in GitHub repository. However, I did not run the code by myself.

Reviewer #2

(Remarks to the Author)

I thank the authors for addressing my final concerns and I have no further comments.

(Remarks on code availability)

Reviewer #3

(Remarks to the Author)

(Remarks on code availability)

Point-by-point rebuttal to reviewers' comments

Reviewer #1 (Remarks to the Author):

Kuzub and colleagues used 1321 viral metagenomes of infant gut and 55 negative controls (NCs) from four public studies to address the viral contamination that are potentially present in the biological samples. The viral analysis pipeline is standard and accepted currently, and they have characterized the viral composition from biological samples and NCs, and then made comparisons at species and strain level. The main benefit of the current study is to provide the database of vOTU sequences identified in NCs that is publicly available for other studies. Beyond that, I personally think that the novelty of this study is limited and makes little advances to the field of early life gut microbiome, and some conclusions are too arbitrary.

We thank the reviewer for their valuable feedback. Please see the replies to other questions below.

Major comments:

1. Line 56-58: the author concluded that NCs did not differ from biological samples in any major genomic and ecological features, including the richness of viral sequences or vOTUs. However, we can see ten times differences in the number (64-594 vs. 0-33). I think this is largely due to the viral variations among biological samples from infants from birth to five year. It is known that the viral richness increases significantly in early life, although the linear mixed-effects model was used for the statistical analyses in order to account the repeated measurement. The quite low number of NCs may also lead to the non-significant results. Therefore, how/what criteria did the author choose these four studies?

We agree with the reviewer that the low number of NCs may have contributed to the non-significant results we observed when comparing viral richness between NCs and biological samples, despite the differences in interquartile ranges. In the present study, we included datasets that met the following criteria (as of January 2024):

- (1) focused on the early-life infant gut virome,
- (2) employed VLP sequencing,

(3) made sequencing data publicly available, and

(4) included NCs in the study design and deposited their sequencing data.

Based on these criteria, we selected four studies from 16 early-life virome studies (see Rebuttal Table 1). We have now updated the description of the study selection in the Methods section; please see lines 296-299:

“To explore the impact of contamination on early-life gut virome studies, we selected studies that: (1) were focused on the early-life infant gut virome, (2) employed VLP sequencing, and (3) had publicly available sequencing data, including NC sequencing data, as of January 2024.”

We now also discuss the potential reasons for the low number of available NCs in our interpretation of the results in the Discussion (lines 257-261): *“The viral richness tended to be lower in NCs compared to biological samples, but this difference was not statistically significant. This lack of significance may be attributed to large variation across studies due to differences in virome extraction, nucleic acid processing, and sequencing methodologies, as well as the overall disparity in the number of NCs and biological samples.”*

Study	Infant samples available	VLP available	NCs reported	NCs deposited	Total number of samples
Kramna et al., 2015 ¹	Yes	Yes	No	No	38
Reyes et al., 2015 ²	Yes	Yes	No	No	170
Lim et al., 2016 ³	Yes	Yes	No	No	48
Zhao et al., 2017 ⁴	Yes	Yes	No	No	220
McCann et al., 2018 ⁵	Yes	Yes	No	No	20
Pannaraj et al., 2018 ⁶	Yes	Yes	No	No	10
Aiemjoy et al., 2019 ⁷	Yes	Yes	No	No	29

Maqsood et al., 2019 ⁸	Yes	Yes	Yes	Yes	78
Yinda et al., 2019 ⁹	Yes	Yes	No	No	24
Liang et al., 2020 ¹⁰	Yes	Yes	Yes	Yes	206
Beller et al., 2022 ¹¹	Yes	Yes	Yes	No	304
Kaelin et al., 2022 ¹²	Yes	Yes	No	No	138
Li et al., 2023 ¹³	Yes	Yes	No	No	45
Shah et al., 2023 ¹⁴	Yes	Yes	Yes	Yes	647
Walters et al., 2023 ¹⁵	Yes	Yes	Yes	No	648
Garmaeva et al., 2024 ¹⁶	Yes	Yes	Yes	Yes	205

Rebuttal Table 1. VLP-based early life virome studies published as of January 2024. Studies that included NCs in the study design but did not have NC sequences deposited are highlighted in yellow. Studies selected for our analyses are highlighted in green.

2. Line 76-77: the author concluded that the viruses identified in NCs resemble the viruses of the human gut, did the author calculate the beta diversity between NCs and biological samples? Based on figure 1e, there are more viral taxa in biological samples than that of NCs.

We agree that the visual representation in Figure 1e is confusing. The y-axis represents the mean fraction of host genus richness across all the samples in a group (NCs versus biological samples), meaning that groups with more samples appear more diverse. As the number of samples greatly exceeds the number of NCs in each study, the biological sample group naturally has a greater number of bacterial host genera.

Based on the reviewers' comments, we calculated the beta diversity between NCs and biological samples and compared it to the beta diversity between unrelated biological samples (Supplementary Figure 6, also pasted below). We have now updated the text to include the results of this analysis (lines 97-106):

“We next assessed the similarity in composition between biological samples and NCs at the vOTU level (1 – Bray-Curtis dissimilarity, Methods). Overall, unrelated biological samples showed a high degree of individual specificity and low inter-similarity, in agreement with previous studies¹⁴⁻¹⁶ (Supplementary Figure 6). In three of the four studies, the similarity between unrelated samples was greater than the similarity between samples and NCs (p -value < 0.0004). In the remaining study, biological samples were more similar to NCs than to other biological samples (Supplementary Figure 6, p -value < 0.0004, Supplementary Data 12). Overall, the similarity of NC composition to that of biological samples was low but comparable to that between unrelated biological samples.”

Additionally, following the reviewer’s comments, we have replaced Figure 1e with a visualization of sample beta diversity using a non-metric multidimensional scaling (NMDS) based on Bray-Curtis dissimilarity, now shown in Figure 1c (revised version, also pasted below). This NMDS analysis demonstrates that most NCs cluster primarily with infant samples. A similar analysis using vOTU aggregates based on the assigned host genus (Figure 1d, revised version, also pasted below) confirms that NCs still cluster with corresponding samples.

In line with these changes, we have also updated the text:

Results section (lines 106-110): *“We also observed NCs clustering with the samples at both vOTU level and the level of host-based vOTU aggregates (Figures 1c-d, Supplementary Data 13). Notably, most of the NCs clustered with infant samples collected during the first four months of life, indicating that the early-life human gut virome exhibits similarity to the NC virome.”*

Methods section (lines 499-503): *“Beta diversity analysis was performed at the vOTU and host-based vOTU aggregate levels using Bray-Curtis dissimilarity. The Bray-Curtis dissimilarity between samples was calculated using the function `vegdist()` from R package `vegan` (v2.6-4). Visualization of sample beta diversity was performed with non-metric multidimensional scaling using the function `metaMDS()` from R package `vegan` with two dimensions.”*

Comparison of vOTU-level similarity index between NCs and samples versus between unrelated samples

Non-metric multidimensional scaling analysis utilizing Bray-Curtis dissimilarity, computed at the vOTU level.

Non-metric multidimensional scaling analysis utilizing Bray-Curtis dissimilarity, computed at the predicted host level.

3. Line 110-113: This is interesting finding and may be useful for studies without setting NCs. However, I think setting own NCs in study is more important, and with the reduce in sequencing cost, setting a few NCs will be easy and valid for own findings.

We agree that including NCs alongside samples is essential to mitigate and understand the impact of possible contamination, as emphasized in lines 288–293 of the manuscript: *“Given our results regarding the higher susceptibility of early-life virome samples to contamination, we emphasize that NCs from multiple sources and processing stages must be included in future virome studies. We anticipate that public data-sharing of NCs from such studies will enhance the NC database and improve the quality of early-life gut virome data for future reuse.”*

However, considering that many published datasets lack deposited NCs (see the Rebuttal Table 1 attached to our response to comment #1), we believe our resource remains valuable for enhancing data reusability by providing a way to estimate approximate levels of contamination of biological samples. Adjustment for these levels might be crucial for a proper control for Type I errors in testing a range of hypotheses in virome-wide studies, e.g. statistical tests involving alpha, beta and gamma diversity, as well as other scenarios.

4. Line 124-126: the proportion of vOTUs shared with NCs was significantly higher in infants than in mothers. As we know, the richness of viruses in infants is much lower than that of adults, which may lead to higher proportion of viral contamination. Thus, it is hard to draw that conclusion that lab-induced contamination is more likely to be sequenced and detected in early-life gut virome samples.

We agree that proportional data can be hard to interpret as infants have lower viral richness compared to adults. We have thus revised Supplementary Figure 4b using calculations based on absolute numbers (see Figure 3b, pasted below). As shown, the number of vOTUs shared with NCs remains higher in infant samples compared to maternal samples. Both figures support our conclusion that early-life VLP samples share more vOTUs with NCs and are therefore more susceptible to contamination.

Number of vOTUs shared 611 with NCs from the same study in maternal samples compared to the infant samples.

5. Line 133-135: may I ask how figure 2f reflect the finding here?

We agree that Figure 2f does not reflect the notion of the proportion of vOTUs shared between NCs and biological samples. We have moved the reference to Figure 2f (Figure 3g in the revised manuscript) to the appropriate place (lines 207-212), where the reconstructed ultrametric tree for *Bacteroides* phage L6428 supports the distinction between samples that share identical strains with NCs and those that harbour distinct strains. To clarify this distinction, we have now added coloured rectangles around the clades that contain strains identical to those found in NCs and updated legends of Figures 2 and 3 as follows (lines 598-599, 621-622): “Cases of identical strain sharing between NCs and samples are highlighted in pink rectangles”.

6. Line 156-164: the overall conclusion is quite general and these are the practical tips for sequencing, such as contaminations may come from multiple sources at different processing stages, and setting controls are very necessary. Additionally, “infant feces sampled closer to birth are more prone to contamination and share more vOTUs with NCs than feces sampled later in life.” is needed to reanalyzed as commented above. As shown in Rebuttal Table 1, we provided in response to the reviewer’s comment #1, only a few studies have reported NC sequencing, and even fewer have made their NC data publicly available. In the first scenario, the lack of NCs risks rendering study results inaccurate if contamination is present. In the second scenario, the lack of publicly available NC data makes data reuse challenging.

Given these points, even if our conclusions appear general, they are not obvious and do not reflect current standard practices in the field. We thus feel it is critical to emphasize the importance of NC inclusion in study design and the processing of NCs alongside samples to account for contamination. We also emphasize that this need is even more pronounced in early-life studies VLP studies (lines 288-291), where samples are prone to contamination due to their low biomass. As can be seen from our response to the reviewer's comment #4, this conclusion remains valid even when analysing absolute numbers of vOTUs shared with NCs.

Minor comments:

7. Line 136: how is the 71.5% obtained? Is it not based on 806/1321?

We thank the reviewer for pointing out the miscalculation. It has been corrected in the text, which now reads: *“Across the entire dataset, 71.5% (N=944) of samples share at least one vOTU with the NCs of their own study, and 85.4% (N=806, 61.0% of the total number of samples) of those shared at least one strain identical to one detected in NCs.”*

8. Line 94-95: Figure 2c is hard to understand.

We acknowledge that the y-axis title is hard to understand and have now changed it to “log10 similarity index” and added additional explanation to the Figure 2c legend (Figure 2d in the revised manuscript, lines 599-602): *“vOTU level compositional similarity between NCs (left panel) and NCs vs. samples (right panel) within and across studies. Each dot represents the similarity index between two samples, calculated as $1 - \text{Bray-Curtis dissimilarity index}$ (Y-axis is log10-transformed).”*

We have also revised the vertical panel titles to “NCs vs NCs” and “NCs vs Samples” for clarity and added “study” as the X-axis title.

9. The meaning of asterisk should be provided in all legends.

We have clarified the meaning of asterisks and added the following text to the legends: Figure 1 legend (lines 586-589): *“Asterisks denote statistical significance: **FDR < 0.010, ns=not significant. 'NA' is used when significance cannot be calculated.”*

Figure 2 legend (lines 604-605): “Asterisks denote statistical significance: ***FDR < 0.001”

Figure 3 legend (lines 625-626): “Asterisks denote statistical significance: ***FDR < 0.001, ****FDR < 0.0001, ns=not significant.”

10. Figure 2f: what is meaning with “shah_kid122_N774_L6428_K6.1_E0_P0_F0”?

In Figure 2a “shah_kid122_N774_L6428_K6.1_E0_P0_F0” and “maqsood_C0251iv_N4_L41225_K60.7_E0_P0_F0” refer to the unique contig IDs of putative phage genomes reconstructed in our dataset. We acknowledge that this information was not mentioned in the text. To address this, we have renamed the vOTU representatives in Figure 2a (Figure 2b in the revised manuscript) to “Burkholderia phage L41225” and Figure 2f (Figure 3g in the revised manuscript) to “Bacteroides phage L6428”, where ‘L’ followed by a number represents the contig length. We have also added the following explanation of the sequence naming to the caption for figure legends:

Figure 2 legend (lines 595-596): “**b.** *the Burkholderia phage L41225 (‘L’ followed by a number represents the genome length)*”;

Figure 3 legend (lines 618-619): “**g.** *Reconstructed ultrametric tree for a Bacteroides phage L6428 (‘L’ followed by a number represents the genome length).*”

Reviewer #1 (Remarks on code availability):

The code is clear.

Reviewer #2 (Remarks to the Author):

In this study, Kuzub et al. investigate the impact of environmental contamination on virome datasets from human gut studies. Using publicly available virome data from infant cohorts, the authors compared biological samples with negative controls (NCs) to identify vOTUs that were shared between the two types of samples. Such sharing events would then imply possible contamination. They found that while most biological samples indeed shared at least one vOTU with the NCs, these viruses most commonly differed at the strain-level, making them less likely to reflect an actual contamination event. The authors thus recommend performing virome decontamination at the strain

level rather than broader taxonomic levels, and also show that even when NCs are unavailable, decontamination against NCs from other studies is nonetheless useful.

Overall, I found the study to be interesting and valuable, emphasizing the importance of estimating and managing contamination in virome datasets. However, I found the authors' results and conclusions to be somewhat inconsistent, and the impact of contamination appears overstated. For example, the authors claim that 71.5% of samples contain at least one contaminant. But, after decontamination, the authors show this only reduces 1.5% of overall richness, which suggests these contaminants represent a very small portion of the sample diversity. It is important to include additional analyses to better understand the impact of the contamination identified on any subsequent biological conclusions. I provide below more specific comments:

We thank the reviewer for their comments and suggestions and have revised the manuscript in line with their comments. Please see our replies below for details.

1. Though the various analyses at vOTU- and at strain-level are well-described, the bottom-line recommendations for other researchers when performing decontamination should be sharpened and justified. Specifically, given that one of the main messages of the study is that decontamination should be done at the strain-level rather than the vOTU level, it is unclear why the authors nonetheless offer readers an NC-associated vOTU catalog to estimate contamination (lines 110-113), and conclude the main text with a recommendation to use vOTUs (156-160). If this is due to technical considerations (e.g., hurdles of strain-level analysis), and vOTU sharing is a good-enough proxy, this argument should be made in the paper.

We thank the reviewer for their comment and agree that the use cases for a) strain-level decontamination and b) contamination estimation using the NC vOTU catalog should be clarified. Briefly, whenever NCs are included in the study design and their sequencing data is available, strain-level decontamination has to be utilized. When NCs were not included in the study design, especially in VLP-enriched low-biomass datasets, we suggest estimating contamination using the NC-associated vOTU catalog and adding the percentage of richness shared with the catalog as a correction factor in downstream analyses. Environmental contamination estimated with this catalog shows concordance with the percentage of strain-level sharing between

biological samples and within-study NCs (Figure 4b in the revised manuscript, also pasted below). We have, therefore, clarified the differences in decontamination strategies for studies with internally generated NCs versus external NCs in our Methods, Results, and Discussion. The updated manuscript text now reads:

Methods section (lines 437-456): *“To assess the feasibility of using the NC-associated vOTU-based catalog²¹ to estimate contamination in the absence of internal NCs, we compared two metrics: (1) the percentage of reads from the biological samples that mapped to the catalog and (2) the percentage of vOTUs shared with the catalog per biological sample. We evaluated the reliability of these metrics by assessing their concordance with the percentage of sample richness shared with the studies’ internal NCs at strain level. In the first approach, for each study, we created a sub-database of vOTUs detected in the NCs of the other three studies and mapped reads from biological samples to these vOTUs using Bowtie v.2.5.1 with --very-sensitive flag. The contamination metric was calculated as the proportion of reads mapped to contigs with more than 50% or 75% breadth coverage. This approach showed low to moderate concordance with the percentage of sample richness shared with studies’ internal NCs at strain level ($0.11 \leq \rho \leq 0.57$, Spearman correlation), likely due to spurious mapping or artificially inflated mapping coverage due to multimapping artifacts. In the second approach, we calculated the percentage of vOTUs shared per biological sample with the catalog using the vOTU table. Specifically, for each study, we determined the percentage of richness in a biological sample that overlapped with NCs from the other three studies. This approach demonstrated moderate to high concordance with the percentage of sample richness shared with studies’ internal NCs at the strain level for three out of four studies ($0.34 \leq \rho \leq 0.82$, Spearman correlation, Figure 4b).”*

Results section (lines 233-245): *“Given the limited amount of strain-sharing with external NCs, we tested if the proportion of vOTUs shared with external NCs could provide an estimate of contamination for studies where NCs are not available and direct decontamination is not possible. Of all the biological samples, 41.1% of samples shared a median of 0.7% (IQR: 0.2–4.8) of vOTUs per sample with NCs from other studies, although this vOTU-sharing was significantly lower than that with own NCs in two out of four studies (Supplementary Data 22, Supplementary Figure 9a). The proportion of vOTUs shared with external NCs showed moderate to high correlation*

to the proportion of strains shared with own NCs ($0.34 \leq \rho \leq 0.82$, Figure 4b, Supplementary Data 23). We also tested if a proportion of the reads mapped to the genome sequences of vOTU representatives identified in external NCs could be used to estimate contamination, but this provided lower concordance ($0.11 \leq \rho \leq 0.57$, Supplementary Figure 9b, Supplementary Data 24).

These results suggest that sample contamination could be estimated using the proportion of vOTU-sharing between samples and NCs from independent studies, and this estimate could potentially be used as a correction factor in downstream statistical analysis. To facilitate the use of available NCs in quality control of future studies, we have made the database of vOTU sequences identified in NCs publicly available¹⁴.

Discussion section (lines 281-284): *“While this method cannot replace direct decontamination with internal NCs, the estimate of contamination levels it offers can support sample quality control and the inclusion of contamination as a correction factor in statistical analyses.”*

Correlation of the percentage of strains shared between samples and internal NCs with the percentage of vOTUs shared between samples and NCs from external studies

2. I agree with the premise that strain-level analysis does provide more accurate decontamination procedures. However, the authors need to expand their analyses to better support how much of an impact this actually makes. For example, how does their decontamination procedure change the results of the original publications cited, if at all? As mentioned above, if removing contamination only filters out 1.5% of the sample diversity, then I am not sure how biologically meaningful this whole procedure is in practice.

We thank the reviewer for their comment. While the median percentage of sample richness represented by strains identical to NCs is low, for 12% of infant samples (N=137) at least 10% of their virus richness was represented by NC-identical strains. This percentage varied depending on the study. In a similar percentage of infant samples (11%, N=122), the abundance of contaminants exceeded 10%. Given that >10% of samples in both cases were affected by contamination and that VLP studies often rely on a limited number of biological samples, it is crucial to estimate and address contamination levels in early-life samples to ensure accurate data interpretation and reliable decontamination. We have now added Figures 3 d-e (also pasted below) to visualize these findings.

Given these results, we have updated the text accordingly. It now reads (lines 198-200): *“While the median number of strains shared between biological samples and NCs was 2 (IQR: 0–28), the abundance of contaminants exceeded 10% in 11% of infant samples (Figure 3e).”*

Study-wise distribution of infant samples categorized by the percentage of richness of strains shared with NCs. The dashed blue line indicates 10% of samples.

Study-wise distribution of infant samples categorized by the abundance of strains shared with NCs. The dashed blue line indicates 10% of samples.

3. The results are presented in a way that appears to overstate the extent of contamination discovered. Upon closer inspection, the differences are much more subtle. For instance, the authors use statistics such as number of vOTUs, contigs, etc. to argue that biological samples and NCs are indistinguishable. However, in Fig. 2c, where the sample composition is taken into account, the median similarity observed between samples and NCs appears to be close to 0 for all the comparisons. This implies that at a compositional level, biological samples and NCs are in fact very distinct. The authors need to be more upfront about this. I suggest performing a PCoA analysis per study and include a figure showing how biological samples and NCs cluster in a 2D space.

We thank the reviewer for their comment and suggestions, which have helped us to improve the manuscript. We agree that compositional differences between NCs and biological samples have to be tested and explicitly acknowledged. We have therefore updated the text, which now reads (lines 97-106): *“We next assessed the similarity in composition between biological samples and NCs at the vOTU level (1 – Bray-Curtis dissimilarity, Methods). Overall, unrelated biological samples showed a high degree of individual specificity and low inter-similarity, in agreement with previous studies¹⁴⁻¹⁶*

(Supplementary Figure 6). In three of the four studies, the similarity between unrelated samples was greater than the similarity between samples and NCs (p -value < 0.0004). In the remaining study, biological samples were more similar to NCs than to other biological samples (Supplementary Figure 6, p -value < 0.0004 , Supplementary Data 12). Overall, the similarity of NC composition to that of biological samples was low but comparable to that between unrelated biological samples.”

Following the reviewer’s suggestion, we also made an NMDS plot based on Bray-Curtis dissimilarity (see Figure 1c, also pasted below). The results show that most of the NCs cluster with samples. Additionally, we observe that most NCs cluster with infant samples collected from birth to 4 months postpartum. This observation implies that at least a subset of the samples share a significant number of vOTUs with the NCs, which could potentially be attributed to contamination.

Non-metric multidimensional scaling analysis utilizing Bray-Curtis dissimilarity, computed at the vOTU level.

4. Related with the above point, most of the results are presented in boxplots that are log-scaled (even after relative abundance normalization). This gives a misleading

impression that the distribution values are much higher. I suggest the authors to not use log-transformation at least when presenting differences in terms of proportions (e.g., Fig. 2e and Supplementary Fig. 4b).

We agree and have now changed Figure 2e (Figure 3c in the revised manuscript, also pasted below) and Supplementary Figure 4b (Figure 3a in the revised manuscript, also pasted below). We have kept the log scale for the other figures (Figures 1a, 1b, 2d; Supplementary Figures 2a-b, 3a-b, 6, 8 in the revised manuscript) that have a large between-sample variation and outliers that make the distribution of data points unreadable.

Percentage of sample richness represented by NC-shared vOTUs per infant timepoint

Percentage of sample richness represented by NC-shared vOTUs from the same study in maternal samples compared to infant samples.

5. Furthermore, beyond showing what proportion of vOTUs are shared, it is important to also accurately assess which ones are not differentially abundant between NCs and biological samples using proper statistical tests. The authors should perform a statistical analysis (e.g., using mixed effects models) to identify which vOTUs are or are not statistically different between groups. The results could then be illustrated in a heatmap to present the abundance variation of selected vOTUs within each sample group.

We thank the reviewer for the opportunity to improve our analysis. As suggested, we have now performed a differential abundance analysis comparing NCs to samples and found that the results varied across studies. To ensure robustness, we included only vOTUs detected in at least two samples and two NCs. This led to the exclusion of Garmaeva et al. study from this analysis as it had only one NC. The results of this analysis are summarized in the table below and in Supplementary Figures 8a-c, also pasted below). Since Shah et al. study included 2,914 vOTUs for comparison, we used volcano plots rather than heat maps to illustrate the results of this analysis.

Study	Number of vOTUs with	Number of vOTUs with similar
-------	----------------------	------------------------------

	abundance differing between biological samples and NCs	abundance in biological samples and NCs
Maqsood et al.	0	6
Liang et al.	28	30
Shah et al.	2896	18

Rebuttal Table2. Number of NC-shared vOTUs with differential and similar abundance in samples and NCs.

Additionally, we have updated the manuscript text to reflect these findings. The revised text now reads:

Results section (lines 165-172): *“In two of the three studies tested, the abundance of individual NC-shared vOTUs was consistently higher in NCs compared to samples, with 48.3% (N=28) and 98.9% (N=2,613) of tested NC-shared vOTUs more abundant in NCs in the Liang et al. and Shah et al. studies, respectively (Supplementary Figure 8, Supplementary Data 17). In Maqsood et al. none of the six NC-shared vOTUs we tested were differentially abundant between samples and NCs, and five of these also showed no differences between NCs and samples in the Liang et al. study. For these overlapping vOTUs, Burkholderia bacteria were predicted as hosts.”*

Methods section (lines 533-541): *“For the identification of vOTUs that were differentially abundant between NCs and biological samples, we included only the vOTUs detected in at least two samples and two NCs to ensure robust comparisons. This filtering reduced the number of vOTUs analyzed and led to the exclusion of the Garmaeva et al. study, which had only one NC. The final dataset comprised 6 vOTUs from Maqsood et al., 58 vOTUs from Liang et al., and 2,914 vOTUs from Shah et al. The analysis was done separately for each study using linear mixed models (lme4 and lmerTest) with the subject group included as a random factor (Supplementary Data 17) to assess significance and using log2 fold change to evaluate differences in abundance.”*

Volcano plots for the vOTU differential abundance analysis visualisation between NCs and samples in the studies. Every point corresponds to a vOTU detected in at least 2 samples and 2 NCs in the same study. Significantly enriched vOTUs in NCs are shown

as red dots in the upper right panels, outlined by dashed lines, while those enriched in samples appear as red dots in the upper left panels. Labels represent a unique sample identifier comprised of the assembly node number and sequence length (Supplementary Data 17). The horizontal dashed line represents an FDR threshold of 0.05. The vertical dashed lines denote log-fold change cut-offs of -1 and 1. Significance was calculated using linear-mixed models.

6. The authors should clarify in the Abstract what proportion of total vOTUs detected were contaminants (I assume this is only 3%, 5,984/193,970) and how much of the abundance of each sample on average was composed of contaminant OTUs. In addition, including a figure (e.g., Venn diagram) showing what proportion of vOTUs were found to be unique to biological samples, controls or shared would be very helpful.

We believe that reporting the proportion of vOTUs detected in NCs relative to the total number of all discovered vOTUs may not provide a meaningful interpretation in this case. This is because there are fewer NC samples compared to the total number of biological samples, and every additional NC adds, on average, 132 new vOTUs (linear regression, $\beta=132$, $p\text{-value}=5.6e\text{-}32$).

Increasing number of vOTUs identified in NCs with each added NC.

As suggested, we have now included the Venn diagram (Supplementary Figure 7, pasted below). We have also modified the abstract by adding the median abundance of contaminants in samples sharing at least one identical strain to NCs (lines 15-18): *“We show that 61% of samples share at least one identical strain with NCs, likely representing external contamination. While the median abundance of contaminant strains in these samples is only 1%, it ranges from 0 to 99% and exceeds 10% in 11% of infant samples.”*

Venn diagram for the vOTUs detected in NCs and biological samples.

7. In the main text when first mentioning vOTU creation, it is worth defining that vOTUs roughly reflect species-level clusters, to make it clear that when moving to strain-level you are increasing rather than decreasing resolution.

We have now added this information to the text, please see lines 51-56 of the revised manuscript: *“Together, these studies include 1,321 biological samples (1,175 infant samples from 0 to 5 years and 146 maternal samples) and 55 NCs (Supplementary Data 1, Supplementary Figure 1), in which we identified 971,583 putative virus sequences that clustered to 193,970 viral operational taxonomic units (vOTUs), providing a general representation of species-level viral clusters (see Methods, Supplementary Figure 2a,b).”*

8. The human gut is not considered a low-biomass environment per se. The issue of contamination in the samples here studied relate with the viral enrichment protocols. Important to clarify this.

We agree and have rephrased the text of the revised manuscript to address the reviewer's comment:

Abstract (lines 11-13): *“Contaminant sequences of external origin complicate the study of host-associated viromes, particularly in low-biomass samples obtained through viral-like particle (VLP) enrichment.”*

Introduction (lines 45-48): *“Here, we aimed to identify environmental contaminants using the viral composition in NCs as a proxy in order to assess the impact of environmental contamination on the low-biomass samples obtained using viral-like particle (VLP) enrichment protocols and elucidate the level of genomic resolution necessary for virome decontamination.”*

9. Some sentences should be rephrased to improve readability and clarity. Examples: lines 92-94, 102-104, 104-106, ...

We agree and have rephrased these sentences as follows:

Lines 194-195: *“Furthermore, within each study, 7.7–23.9% of the NC-detected vOTUs were identical to those found in biological samples at strain level.”*

Lines 162-165: *“75.3% of the samples shared vOTUs with NCs from the same study, with a median of 4.9% (IQR: 1.9–12.9) vOTUs per sample. The median abundance of all vOTUs shared with NCs per sample was 1.7% (IQR: 0.3–8.7).”*

Lines 236-239: *“Of all the biological samples, 41.1% of samples shared a median of 0.7% (IQR: 0.2–4.8) of vOTUs per sample with NCs from other studies, although this vOTU-sharing was significantly lower than that with own NCs in two out of four studies (Supplementary Data 22, Supplementary Figure 9a).”*

10. Throughout the manuscript, when noting a p-value, FDR, or another statistic, it should be clear what statistical test was used (e.g., line 48, line 51, etc.).

We agree and have now linked all reported results of statistical tests to the corresponding Supplementary Data table, which contains all relevant details

(statistical test used, p-value, and FDR) each time they appear in the main text. We also added a clarification in the "Statistics and Reproducibility" section of the Methods (lines 474-476): *"For detailed information regarding the tests used to assess the significance of results, see Supplementary Data."*

This section also provides a general description of the tests used throughout the study. While we considered including all the details of each statistical test in the main text, we felt this would negatively impact the readability of the manuscript, so we opted not to add this level of detail in the text itself.

11. Line 77-78: Improve conciseness - "at least at the highest taxonomy and predicted host levels"?

We have removed the sentence about the similarity at the highest taxonomy level of viruses and their predicted hosts, leaving the clearer and more concise sentence in the revised manuscript (lines 93-95):

"At the genus level of prokaryotic viruses' hosts, NCs primarily contained viruses with hosts typical of the human gut microbiota, such as Alistipes, Bacteroides, Bifidobacterium, and Escherichia (Supplementary Figure 5)."

12. In Supplementary Figure 1 the term "vOTUr" is used but not defined anywhere. A typo maybe?

Thank you for noticing this typo. We have corrected it to vOTU.

13. Across several figures, 'NA' is listed instead of significance levels / "ns" when comparing boxplots. If this is on purpose it should be noted in the legend.

We have included the clarification at the end of the legends for all relevant figures:

Figure 1 legend (lines 586-589): *"Asterisks denote statistical significance: **FDR < 0.010, ns=not significant. 'NA' is used when significance cannot be calculated."*

14. Line 155: When you note that your pipeline has been made available, please refer to where readers can find it.

The pipeline can be found at GitHub in the folder: https://github.com/GRONINGEN-MICROBIOME-CENTRE/NCP_VLP_project/. We have also clarified this in the manuscript, see lines 271-273:

“We have also developed a pipeline for strain-level decontamination (https://github.com/GRONINGEN-MICROBIOME-CENTRE/NCP_VLP_project).”

15. The term “compositional similarity”, used in several places in the paper, is slightly ambiguous and can refer to a few different things. Consider replacing it with a more concise term.

We have added the requested specifications in lines 160-162 of the revised manuscript, please see below.

“Only 20.3% (N=256) of all biological samples did not share any vOTUs with any NCs, while some shared up to 100% of their composition at the vOTU level.”

16. The methodology of defining which contigs are viral based on the 4 different tools, as described in lines 232-234, is unclear. Did you mean you chose only contigs that had been labeled as viral by all methods?

For our analysis, we included all sequences identified as viral by at least one of the tools. We agree that the methodology was unclear in the original text and have clarified this in the revised manuscript. Please see lines 366-367 of the revised manuscript: *“All contigs identified as viral by at least one of these tools were selected for further analysis.”*

17. Line 243: The correct term should be “clustering” rather than “dereplication”.

While we appreciate the comment, we respectfully disagree. According to Evans and Denef (2020, *mSphere*)¹⁷, dereplication is defined as the reduction of a set of genomes based on high sequence similarity between these genomes. In this sentence we described the process of reduction of the number of sequences (N=971,583 viral contigs) based on the genome similarity, resulting in the retrieval of the 307,938 vOTU representatives that were subsequently used for the data analysis. As this process aligns with the definition of dereplication above, we prefer to retain the original text.

18. To place the current work in context of other similar studies, it may be interesting to compare the current catalog of contaminating viral sequences to other catalogs published (using a different approach) such as this one: 10.1016/j.cmi.2019.04.028. Alternatively, you may note other approaches for viral decontamination.

We agree with the reviewer and have now compared the assigned taxonomy of vOTUs detected in the NCs in the present study to the taxonomy of laboratory-component-associated (LCA) viral sequences from Asplund et al. (2019)¹⁸. Briefly, vOTU representative sequences of vOTUs identified in NCs (N=5,984) were blasted against the non-redundant nucleotide (NT) database using settings similar to those used in Asplund et al. (2019). In total, 1,463 NC-detected sequences had hits against the NT database (query coverage >30%). Of those, only 33 viruses overlapped with Asplund et al. LCA viral sequences. This analysis is described in detail in the manuscript and reads as follows:

Methods (lines 405-411): *“Sequences identified in the NCs were queried against the NT database (updated 25 November 2023) using blastn search with an e-value of 1×10^{-3} . The best hit was selected based on e-value, query coverage, percent identity, and alignment length. These hits were then compared with the LCA viral sequences reported by Asplund et al. (2019)⁶ to identify overlapping hits. Hits with query coverage < 30% were discarded as potentially spurious matches. NT-hits to viral taxa that did not match LCA are reported in Supplementary Data 15.”*

Results (lines 135-145): *“We also compared the sequences detected in NCs to the genomes of viral taxa reported to be laboratory-component-associated (LCA) viral sequences⁶ by Asplund et al. (2019). Of the 5,984 NC sequences, only 34 (0.6%) matched the previously reported LCA viral sequences (see Methods). Of these, 60.1% (N=20) belonged to Bordetella phage, EBPR podovirus, and sewage-associated circular DNA viruses (Supplementary Data 14). Among the 677 NC sequences with NT (non-redundant nucleotide database) viral taxonomic assignments that did not overlap with the LCA taxa, 529 were linked to metagenomically reconstructed viral genomes for which taxonomy was only resolved up to the class level. Interestingly, within the subset of sequences assigned to isolate viral genomes, we identified a few previously reported reagent-associated CRESS-like viruses²⁰ (Supplementary Data 15).”*

19. A few highly relevant papers seem to be unmentioned and should be referred to in the introduction: 10.3389/fmicb.2021.745076, 10.1016/j.cmi.2019.04.028.

We completely agree and have now incorporated the references and definitions from Asplund et al. 2019 and Jurasz et al. 2021 into the Introduction and Discussion, please see the example below:

Introduction (lines 38-41): “Previous studies have linked the environmental contamination identified in virome samples to various laboratory components used for nucleic acid extraction and sequencing, including individual reagents⁶, entire extraction kits¹², and laboratory plastics⁵.”

Discussion (lines 261-263) “Although the contamination was largely study-specific, we did identify a few previously reported contaminants^{6,20}, but with rather low overlap.”

Reviewer #2 (Remarks on code availability):

I reviewed the above GitHub repository which appears to contain all the relevant code with sufficient documentation. However, I have not run the code myself.

Reviewer #3 (Remarks to the Author):

References

1. Kramná, L. *et al.* Gut Virome Sequencing in Children With Early Islet Autoimmunity. *Diabetes Care* **38**, 930–933 (2015).
2. Reyes, A. *et al.* Viruses in the faecal microbiota of monozygotic twins and their mothers. *Nature* **466**, 334–338 (2010).
3. Lim, E. S. *et al.* Early life dynamics of the human gut virome and bacterial microbiome in infants. *Nat Med* **21**, 1228–1234 (2015).
4. Zhao, G. *et al.* Intestinal virome changes precede autoimmunity in type I diabetes-susceptible children. *Proceedings of the National Academy of Sciences* **114**, E6166–E6175 (2017).
5. McCann, A. *et al.* Viromes of one year old infants reveal the impact of birth mode on microbiome diversity. *PeerJ* **6**, e4694 (2018).
6. Pannaraj, P. S. *et al.* Shared and Distinct Features of Human Milk and Infant Stool Viromes. *Front. Microbiol.* **9**, (2018).
7. Aiemjoy, K. *et al.* Viral species richness and composition in young children with loose or watery stool in Ethiopia. *BMC Infectious Diseases* **19**, 53 (2019).

8. Maqsood, R. *et al.* Discordant transmission of bacteria and viruses from mothers to babies at birth. *Microbiome* **7**, 156 (2019).
9. Yinda, C. K. *et al.* Gut Virome Analysis of Cameroonians Reveals High Diversity of Enteric Viruses, Including Potential Interspecies Transmitted Viruses. *mSphere* **4**, 10.1128/msphere.00585-18 (2019).
10. Liang, G. *et al.* The stepwise assembly of the neonatal virome is modulated by breastfeeding. *Nature* **581**, 470–474 (2020).
11. Beller, L. *et al.* The virota and its transkingdom interactions in the healthy infant gut. *Proceedings of the National Academy of Sciences* **119**, e2114619119 (2022).
12. Kaelin, E. A. *et al.* Longitudinal gut virome analysis identifies specific viral signatures that precede necrotizing enterocolitis onset in preterm infants. *Nat Microbiol* **7**, 653–662 (2022).
13. Li, H. *et al.* Comparison of gut viral communities in children under 5 years old and newborns. *Virology Journal* **20**, 52 (2023).
14. Shah, S. A. *et al.* Expanding known viral diversity in the healthy infant gut. *Nat Microbiol* **8**, 986–998 (2023).
15. Walters, W. A. *et al.* Longitudinal comparison of the developing gut virome in infants and their mothers. *Cell Host & Microbe* **31**, 187-198.e3 (2023).
16. Garmaeva, S. *et al.* Transmission and dynamics of mother-infant gut viruses during pregnancy and early life. *Nat Commun* **15**, 1945 (2024).
17. Evans, J. T. & Deneff, V. J. To DerePLICATE or Not To DerePLICATE? *mSphere* **5**, 10.1128/msphere.00971-19 (2020).
18. Asplund, M. *et al.* Contaminating viral sequences in high-throughput sequencing viromics: a linkage study of 700 sequencing libraries. *Clinical Microbiology and Infection* **25**, 1277–1285 (2019).

Point-by-point rebuttal to reviewers' comments

Reviewer #1 (Remarks to the Author):

The authors have done additional analyses to improve the confidence of the findings. However, the limited novelty and workload of this study are still held. Lots of the findings or conclusions have already been acknowledged or can be predicted.

We thank the reviewer for their valuable feedback. Please, see the detailed replies below in green.

Regarding the conclusion that NCs did not differ from biological samples in any major genomic and ecological features, including the richness of viral sequences or vOTUs, it is predicted that this non-significance is due to the viral variations among combined biological samples from infants covering the first five year of life and adults (mothers). The very low number of NCs in each study or study-specific variations in their viral composition also make the statistics non-significant.

We thank the reviewer for their valuable feedback. We agree that the lack of statistical significance and consistency may stem from the limited number of negative controls (NCs), the heterogeneous sampling ages of infants across studies, and the inclusion of maternal samples in some datasets. Although we cannot increase the number of NCs due to the limited availability of studies meeting our selection criteria, we have revisited our statistical analyses to address the reviewer's concerns regarding age variation and study-specific effects, as outlined below.

We accounted for study-specific variation in samples and NCs using inverse-rank transformation, which helps to reduce the impact of extreme values and makes the data more comparable across different studies. Indeed, we observed statistically significant differences for some features; however, these were inconsistent across studies for most of the features tested. We have now updated the Results section (lines 64–77) to: *“While clean reads and Shannon diversity were similar between groups (p -value > 0.2, Supplementary Data 2 and 6, Figure 1a,b, Supplementary Figure 3), the number of reconstructed contigs, viral sequences, viral genome completeness, and*

viral richness were generally lower in NCs compared to biological samples (p -value < 0.04, Supplementary Data 3–5 and 7, Figure 1a,b, Supplementary Figure 3). However, both the direction and significance of associations varied across the studies for all features tested except the number of reconstructed viral sequences. For example, viral richness was significantly different between NCs and biological samples in all three tested studies ($FDR < 1e-02$; Supplementary Data 6), but the direction of this difference was inconsistent (Figure 1b). This inconsistency likely reflects both study-to-study variability in vOTU richness (intraclass correlation coefficient (ICC) = 0.2, Supplementary Data 8) and large variation among biological samples (median: 230 vOTUs, interquartile range (IQR): 64–594).”

Our primary aim, however, was to assess whether NCs can be reliably distinguished from early-life biological samples based on the ecological and technical features, without prior knowledge of sample origin. To support this, we conducted an additional analysis that is now described in both the Methods and Results sections.

Methods section (lines 522–529): *“To assess whether genomic and ecological features can distinguish NCs from biological samples based on their hypothetical “sampling age,” we used data from the longitudinal studies by Garmaeva et al. and Liang et al. We constructed a linear model with age (in months) as the outcome variable, using only infant biological samples. For the Liang et al. dataset, we only included samples collected at months 0, 1, 3, and 4. The predictors included the number of reconstructed contigs, discovered viral sequences, genome completeness, and viral richness. All predictor features were inverse-rank transformed prior to model fitting.”*

Results section (lines 78-91): *“Next, we tested whether NCs could be distinguished from biological samples without prior knowledge using the genomic and ecological features that differed between NCs and biological samples (i.e., the number of reconstructed contigs and viral sequences, genome completeness, and richness). We hypothesized that if these features reliably capture biological signals, then a model constructed based on them to predict a quantifiable variable – sampling age – would assign implausible or outlier-like ages to NCs, which by definition do not have a biological age. To test this, we built a linear model to predict sampling age from these four features in two longitudinal studies. In the Liang et al. dataset, the inferred*

sampling age of NCs averaged 1.3 ± 0.6 months, which falls within the observed range for biological samples. Similarly, for the Garmaeva et al. dataset, the NC was assigned an age of 7.9 months, which is also within the dataset's observed range (1–12 months). These results suggest that the combined technical and ecological features, while differing in group comparisons, do not provide a strong enough signal to reliably separate NCs from biological samples.”

As demonstrated by this analysis, without prior knowledge, NCs cannot be distinguished from early-life biological samples using key genomic and ecological features. In light of this, we have revised the title of the corresponding Results section to more clearly convey this conclusion: *“NCs and samples cannot be reliably distinguished using key genomic and ecological features.”*

We have also updated the respective concluding statements in both the Results and Discussion sections for clarity and emphasis:

Results section (lines 124–128): *“Although we observed differences between NCs and biological samples in some genomic and ecological features, the direction and significance were often inconsistent across the studies. Additionally, we demonstrated that NCs could not be reliably distinguished from biological samples based on genomic and ecological features.”*

Discussion section (lines 266–274): *“We also showed that biological samples and NCs did not differ in several technical and ecological features, including the number of clean reads and overall diversity. However, features like viral richness and the number of discovered viruses were generally lower in NCs. Despite these differences, infant samples could not be reliably distinguished from NCs using predictive models based on the combined ecological and technical features, echoing findings from bacteriome studies that compared the compositions of NCs and infant meconium samples²⁴. This suggests that, while certain metrics capture differences between NCs and biological samples, they are insufficient for robust classification.”*

Finally, we also acknowledge all the reviewer’s points regarding study limitations and emphasize in the Discussion section (lines 274-279): *“Moreover, the significance and*

direction of feature-based differences often varied across studies, likely reflecting the large variation across studies due to differences in virome extraction, nucleic acid processing, and sequencing methodologies, as well as the overall disparity in the number of NCs and biological samples and age-related variation in the composition and diversity of biological samples.”

Secondly, the author found that early-life VLP samples share more vOTUs with NCs and are therefore more susceptible to contamination. Are there any biological reasons behind this phenomenon? If it is simply caused by the low diversity of virome in early life, this is easy to understand and has been recognized for all low-biomass samples, and thereby the findings here may make no advances to our current understanding.

We agree that the increased sharing of vOTUs with NCs in early-life samples is mainly due to the low viral load and diversity of the infant gut virome. We have, therefore, added that notion to the Discussion section (lines 262-265): “This increased susceptibility to contamination in infant virome samples is likely linked to their low viral load³ and diversity⁴ and corresponds with earlier findings linking the contamination impact and sample biomass in other metagenomic samples^{5,22,23}”. However, while the relationship between low diversity and contamination risk may seem intuitive, it is often overlooked in practice. Indeed, 12 out of 16 early-life gut VLP virome studies did not sequence or publish NCs alongside their biological samples (see Table 1 below). This oversight is concerning, especially as virome research expands into other environments such as the vaginal and breast milk viromes (Pannaraj et al., 2018¹, Jakobsen et al., 2020²), and the gut viromes of preterm infants (Kaelin et al., 2022³), where one could argue that VLP loads are expected to be even lower. Yet none of the studies mentioned have published NCs. Our study highlights this critical gap and highlights the importance of the inclusion of NCs in the study design.

Study	Infant samples available	VLP available	NCs reported	NCs deposited	Total number of samples
Kramna et al., 2015 ⁴	Yes	Yes	No	No	38
Reyes et al., 2015 ⁵	Yes	Yes	No	No	170
Lim et al., 2015 ⁶	Yes	Yes	No	No	48
Zhao et al., 2017 ⁷	Yes	Yes	No	No	220
McCann et al., 2018 ⁸	Yes	Yes	No	No	20
Pannaraj et al., 2018 ¹	Yes	Yes	No	No	10
Aiemjoy et al., 2019 ⁹	Yes	Yes	No	No	29
Maqsood et al., 2019 ¹⁰	Yes	Yes	Yes	Yes	78
Yinda et al., 2019 ¹¹	Yes	Yes	No	No	24
Liang et al., 2020 ¹²	Yes	Yes	Yes	Yes	206
Beller et al., 2022 ¹³	Yes	Yes	Yes	No	304
Kaelin et al., 2022 ³	Yes	Yes	No	No	138
Li et al., 2023 ¹⁴	Yes	Yes	No	No	45
Shah et al., 2023 ¹⁵	Yes	Yes	Yes	Yes	647
Walters et al., 2023 ¹⁶	Yes	Yes	Yes	No	648
Garmaeva et al., 2024 ¹⁷	Yes	Yes	Yes	Yes	205

Rebuttal Table 1. VLP-based early life virome studies published as of January 2024. Studies that included NCs in the study design but did not have NC sequences

deposited are highlighted in yellow. Studies selected for our analyses are highlighted in green.

Finally, contamination has been frequently discussed and concerned during sample preparation and sequencing. Therefore, it is still recommended to set own NCs in each study, and with the reduce in sequencing cost, setting a few NCs will be easy and valid for own findings, instead of downloading these deposited sequences and blasting reads to them, which makes the situation more complicated. Besides, two studies (out of four studies in total) show very low correlations when calculating the proportion of vOTUs shared with external or own NCs.

We completely agree that including multiple internal NCs is the most valid and reliable approach and have emphasized this point in the Discussion (lines 317-322): *“Given our results regarding the higher susceptibility of early-life virome samples to contamination, we emphasize that NCs from multiple sources and processing stages must be included in future virome studies.”*

However, when reusing data from previously published studies where internal NCs are unavailable, the method we describe remains one of the few viable approaches for addressing contamination. We believe this is particularly valuable for the early-life gut research community, given the difficulties in generating VLP datasets and active reuse of existing virome datasets. For example, while recent studies based on the combination of virome datasets extended the diversity of metagenomically assembled viruses and offered new virus catalogs, such as MAGIC¹⁸ and ELGV¹⁹, these studies did not document any environmental contamination filtering, most likely because NC data were not available for most of the reused studies, making it difficult to account for contamination.

We highlight this in the Discussion (lines 303-313) as well, emphasizing that our method is specifically intended for data reuse scenarios and cannot replace the use of study-specific NCs. *“We acknowledge the existence of scenarios when proper NCs are not available, such as the reuse of previously generated datasets. <...> Therefore, vOTU-sharing with the NC-associated vOTU catalog from this study provides a rough estimate of the general sample contamination level. While this method cannot replace*

direct decontamination with internal NCs, the estimate of contamination levels it offers can support sample quality control and the inclusion of contamination as a correction factor in statistical analyses.”

Line 215-218: why the number of vOTUs shared with own NCs decreased by 33.3%, not 100% after performing strain-level decontamination?

Since vOTUs represent species-level clusters, their sharedness with NCs doesn't necessarily indicate contamination, as demonstrated in lines 210–219 and Figure 3g. Our strain-level decontamination approach distinguishes between cases where samples and NCs share identical strains — classified as contaminants — and cases where the strains are distinct, which are more likely to represent true biological signals from the infant's gut. Consequently, we do not expect a 100% drop in NC-shared vOTUs following decontamination. We now clarified it in the text as follows (lines 220–222): *“After performing strain-level decontamination, the richness of vOTUs, which represents species-level resolution, dropped by an average 1.5% (IQR: 0.5–5.0) in samples that shared vOTUs with NCs.”*

Reviewer #1 (Remarks on code availability):

I have reviewed the code in GitHub repository. However, I did not run the code by myself.

Reviewer #2 (Remarks to the Author):

I have reviewed the revised manuscript and the authors' detailed responses to my comments. In general, I am pleased with the improvements of the manuscript and the thorough revisions made. I have only two remaining comments/concerns:

We thank the reviewer for their valuable comments. Our responses can be found below in green.

1) It would be better to show the new Venn diagram provided on a per study level (Supplementary Figure 7). As the majority of negative control (NC) vOTUs were obtained from the dataset of Shah et al. (5,306 out of the total 5,984 vOTUs) it would be beneficial to see the intersection for each study separately.

We fully agree and have modified Supplementary Figure 7 as suggested, see below.

Rebuttal Figure 1. Venn diagrams for the vOTUs detected in NCs and biological samples.

2) Related with the above point, authors should acknowledge that the majority of their NC vOTU catalog is coming from just 8 NCs from Shah et al. This should also be

more thoroughly discussed as one of the limitations of the study. It would also be important to provide more details on how the samples were originally processed in this study as part of the Methods section. Is there a particular reason why this one study has such high levels of contaminant vOTUs? I could not find any information about these controls in the original paper (<https://doi.org/10.1038/s41564-023-01345-7>).

We agree and have added the suggested updates to the manuscript (lines 133-135): *“In total, 5,984 vOTUs were found in NCs from all four studies (Supplementary Figure 7), with the majority (N=5,339) identified in the NCs from Shah et al.”*

We have also discussed the potential reasons why NCs from Shah et al. had a higher number of sequences (lines 284-295): *“It should be noted that 89.2% of the vOTUs detected in NCs originated from a single study by Shah et al., skewing the catalog toward that dataset. This study also had the highest total number of raw and clean reads per sample and the highest number of vOTUs detected, with 5% of them found in NCs, compared to 0.4–2.8% in the other studies. This slightly elevated proportion may be explained by deeper sequencing and by differences in VLP extraction protocols between studies. The latter could be influenced by the pore size of the membrane filter used for bacterial and large particle removal, the number of filtration steps used, the methods used for VLP concentration, variations in nuclease treatment and DNA extraction kits across the four studies, and the use of the multiple displacement amplification (MDA), which is known to increase sequence sensitivity by amplifying both biological signals and contaminants^{5,25}.”*

Reviewer #3 (Remarks to the Author):

References

1. Pannaraj, P. S. *et al.* Shared and Distinct Features of Human Milk and Infant Stool Viromes. *Front. Microbiol.* **9**, (2018).
2. Jakobsen, R. R. *et al.* Characterization of the Vaginal DNA Virome in Health and Dysbiosis. *Viruses* **12**, 1143 (2020).
3. Kaelin, E. A. *et al.* Longitudinal gut virome analysis identifies specific viral signatures that precede necrotizing enterocolitis onset in preterm infants. *Nat Microbiol* **7**, 653–662 (2022).
4. Kramná, L. *et al.* Gut Virome Sequencing in Children With Early Islet Autoimmunity. *Diabetes Care* **38**, 930–933 (2015).
5. Reyes, A. *et al.* Gut DNA viromes of Malawian twins discordant for severe acute malnutrition. *Proceedings of the National Academy of Sciences* **112**, 11941–11946 (2015).
6. Lim, E. S. *et al.* Early life dynamics of the human gut virome and bacterial microbiome in infants. *Nat Med* **21**, 1228–1234 (2015).
7. Zhao, G. *et al.* Intestinal virome changes precede autoimmunity in type I diabetes-susceptible children. *Proceedings of the National Academy of Sciences* **114**, E6166–E6175 (2017).
8. McCann, A. *et al.* Viromes of one year old infants reveal the impact of birth mode on microbiome diversity. *PeerJ* **6**, e4694 (2018).
9. Aiemjoy, K. *et al.* Viral species richness and composition in young children with loose or watery stool in Ethiopia. *BMC Infectious Diseases* **19**, 53 (2019).
10. Maqsood, R. *et al.* Discordant transmission of bacteria and viruses from mothers to babies at birth. *Microbiome* **7**, 156 (2019).
11. Yinda, C. K. *et al.* Gut Virome Analysis of Cameroonians Reveals High Diversity of Enteric Viruses, Including Potential Interspecies Transmitted Viruses. *mSphere* **4**, 10.1128/msphere.00585-18 (2019).
12. Liang, G. *et al.* The stepwise assembly of the neonatal virome is modulated by breastfeeding. *Nature* **581**, 470–474 (2020).
13. Beller, L. *et al.* The virota and its transkingdom interactions in the healthy infant gut. *Proceedings of the National Academy of Sciences* **119**, e2114619119 (2022).
14. Li, H. *et al.* Comparison of gut viral communities in children under 5 years old and newborns. *Virology Journal* **20**, 52 (2023).
15. Shah, S. A. *et al.* Expanding known viral diversity in the healthy infant gut. *Nat Microbiol* **8**, 986–998 (2023).
16. Walters, W. A. *et al.* Longitudinal comparison of the developing gut virome in infants and their mothers. *Cell Host & Microbe* **31**, 187-198.e3 (2023).
17. Garmaeva, S. *et al.* Transmission and dynamics of mother-infant gut viruses during pregnancy and early life. *Nat Commun* **15**, 1945 (2024).
18. Peng, Y. *et al.* A metagenome-assembled genome inventory for children reveals early-life gut bacteriome and virome dynamics. *Cell Host & Microbe* **32**, 2212-2230.e8 (2024).
19. Zeng, S. *et al.* A metagenomic catalog of the early-life human gut virome. *Nat Commun* **15**, 1864 (2024).